# Exploiting pleiotropy to enhance variant discovery with functional false discovery rates

Andrew J. Bass [1] ✉ & Chris Wallace [1,2] ✉

The cost of recruiting participants for genome-wide association studies (GWASs) can limit sample sizes and hinder the discovery of genetic variants. Here we introduce the surrogate functional false discovery rate (sfFDR) framework that integrates summary statistics of related traits to increase power. The sfFDR framework provides estimates of FDR quantities such as the functional local FDR and $q$ value, and uses these estimates to derive a functional $P$ value for type I error rate control and a functional local Bayes' factor for post-GWAS analyses. Compared with a standard analysis, sfFDR substantially increased power (equivalent to a 52% increase in sample size) in a study of obesity-related traits from the UK Biobank and discovered eight additional lead SNPs near genes linked to immune-related responses in a rare disease GWAS of eosinophilic granulomatosis with polyangiitis. Collectively, these results highlight the utility of exploiting related traits in both small and large studies.

Genome-wide association studies (GWASs) provide a wealth of genetic data to understand the etiology of human diseases. In a GWAS, the discovery of genetic variants requires an adequate sample size to represent the population and maximize statistical power. While increasing sample size increases variant discovery, the sample size is often limited by the cost or availability of participants, particularly in the case of low-frequency or rare diseases.

Given such sample size constraints, an alternative approach is to leverage the ubiquitous genetic correlations (referred to as pleiotropy) between related traits to improve variant discovery[1–4]. One strategy is to use GWAS summary statistics of related traits within a conditional false discovery rate (cFDR) framework to increase power[5]. While a standard GWAS analysis aims to control the probability of at least one false discovery (defined as a variant that does not tag a causal variant), the cFDR approach is more liberal in that it controls the expected proportion of false discoveries (that is, the FDR[6]). Previous work on the cFDR has shown a substantial increase in power when incorporating GWAS summary statistics of related traits compared with a standard GWAS[5,7,8]. The cFDR has thus been applied in GWASs to enhance the discovery of variants[9–12]. However, the utility of cFDR approaches is

limited due to computational costs and strict assumptions of independence between related traits. Although there are other general FDR procedures that can integrate informative data[13–16], it is unclear how to appropriately incorporate GWAS summary statistics while accommodating for dependence due to linkage disequilibrium (LD). Therefore, current approaches cannot fully leverage pleiotropy from multiple related traits to increase power. More generally, the familiar guarantees of family-wise error rate (FWER) control has been a barrier to widespread adoption of FDR methods in GWASs, even though the FDR can substantially increase the number of discoveries in genomics[17].

To address these challenges, we develop a novel method that integrates multiple sets of GWAS summary statistics within the functional FDR (fFDR) framework[15]. The fFDR framework was primarily designed for genomic studies and incorporates a single informative variable (for example, epigenetic or per-gene read depth) when constructing FDR quantities of interest, such as the functional $q$ value (a measure of significance in terms of the positive FDR[17,18]) and local FDR (a posterior error probability[19,20]). Our proposed method, surrogate functional FDR (sfFDR), adapts the fFDR to leverage informative data from multiple sets of GWAS summary statistics while accounting for LD. Importantly,

[1]Department of Medicine, University of Cambridge, Cambridge, UK. [2]MRC Biostatistics Unit, University of Cambridge, Cambridge, UK. ✉e-mail: ab3105@cam.ac.uk; cew54@cam.ac.uk

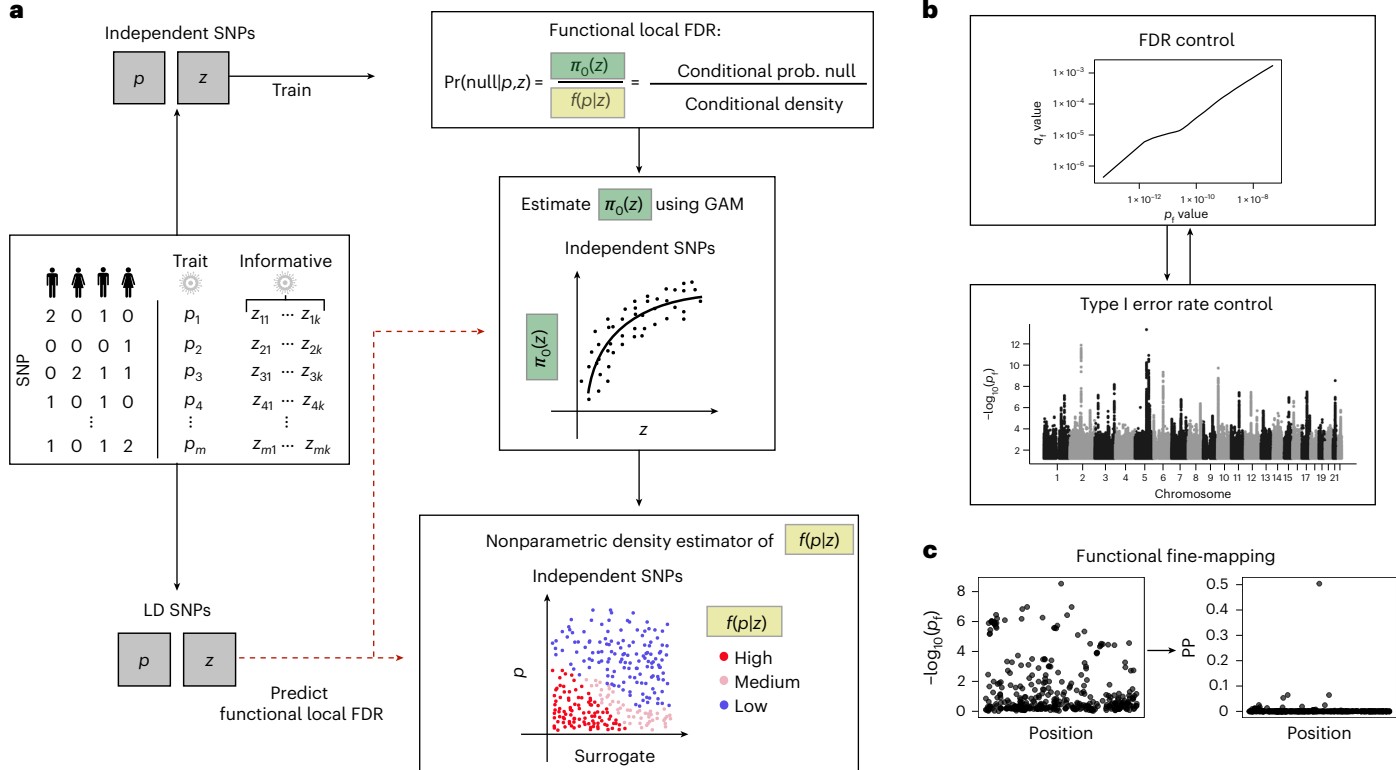

**Fig. 1 | Overview of the sfFDR framework. a**, Estimate the functional local FDR of the primary GWAS *P* values given a set of informative summary statistics. Prob., probability. **b**,**c**, The functional local FDR values are used for estimating the functional *q* value ($q_f$ value) and functional *P* value ($p_f$ value) to control the FDR and FWER, respectively (**b**), and functional fine-mapping (**c**).

sfFDR is a computationally efficient approach and does not assume independence between the related GWAS traits. We also derive a new quantity, the functional *P* value, that incorporates the GWAS summary statistics and can be interpreted like a standard *P* value familiar to GWAS practitioners. Finally, we show how functional local Bayes' factors (BFs) can be calculated from the functional local FDR, allowing a range of post-GWAS analyses to incorporate GWAS summary statistics such as functional fine-mapping and colocalization.

We apply sfFDR to both small- and large-sample GWAS studies to demonstrate its power improvements over a standard GWAS analysis and a joint trait analysis using multitrait analysis of GWAS (MTAG)[21]. We first perform comprehensive simulations to evaluate and compare sfFDR with three general FDR methods extended to our setting. We then demonstrate the power improvements in a study of obesity-related traits from the UK Biobank. Finally, we apply sfFDR to a rare disease GWAS of eosinophilic granulomatosis with polyangiitis (EGPA) and use GWAS summary statistics from related traits (asthma and eosinophil count) to substantially increase power compared with a standard GWAS analysis. We also show how estimates of the functional local FDR can be used to perform functional fine-mapping in the EGPA study, helping to identify the causal locus within a genetic region (assuming a single causal locus).

## Results

### Overview

We briefly review the motivation behind the sfFDR framework (see Methods for additional details). Consider a GWAS for some trait of interest, referred to as the 'primary' GWAS, where a *P* value is calculated on a single nucleotide polymorphism (SNP)-by-SNP basis to assess statistical significance. In a standard analysis, the set of SNPs below a genome-wide significance threshold (for example, $P < 5 \times 10^{-8}$) are classified as statistically significant, with each SNP treated a priori as equally likely to be truly null. However, there is often an abundance of SNP-level information available that can alter our prior belief about whether a SNP is more or less likely to be associated with the trait of interest. In particular, a valuable source of SNP-level information comes from publicly available GWAS summary statistics, where traits with similar genetic architecture can be integrated into the significance analysis to improve power.

Our approach, sfFDR, leverages one or several sets of informative GWAS summary statistics within an extended version of the fFDR framework[15] to improve the power of the primary GWAS (Fig. 1). Given *P* values from the primary GWAS and one or more informative GWAS, **z**, we first identify a LD-independent subset of SNPs. Using the LD-independent SNPs, we estimate the functional local FDR (a posterior error probability), which requires modeling the functional proportion of truly null hypotheses, $\pi_0(\mathbf{z})$, and the conditional density, $f(p|\mathbf{z})$. We estimate $\pi_0(\mathbf{z})$ using a generalized additive model (GAM) and $f(p|\mathbf{z})$ nonparametrically where we use a surrogate variable approximation—the ranked estimated $\pi_0(\mathbf{z})$ values—that circumvents difficulties with higher-dimensional density estimation. The functional local FDR of the left-out dependent SNPs are then predicted from the model fit of $\pi_0(\mathbf{z})$ and $f(p|\mathbf{z})$. With the estimated functional local FDRs, the functional *q* values (referred to throughout as $q_f$ values) are constructed for each SNP and measure significance in terms of the positive FDR (pFDR; closely related to FDR[20]). Intuitively, the $q_f$ value is the minimum probability that a SNP is null given that it is classified as statistically significant (also known as the Bayesian posterior type I error[20]).

The FDR quantities estimated by the sfFDR framework support a range of analyses for GWAS data. In particular, we use the FDR quantities to derive a functional *P* value (referred to throughout as the $p_f$ value), allowing practitioners to control the FWER while incorporating

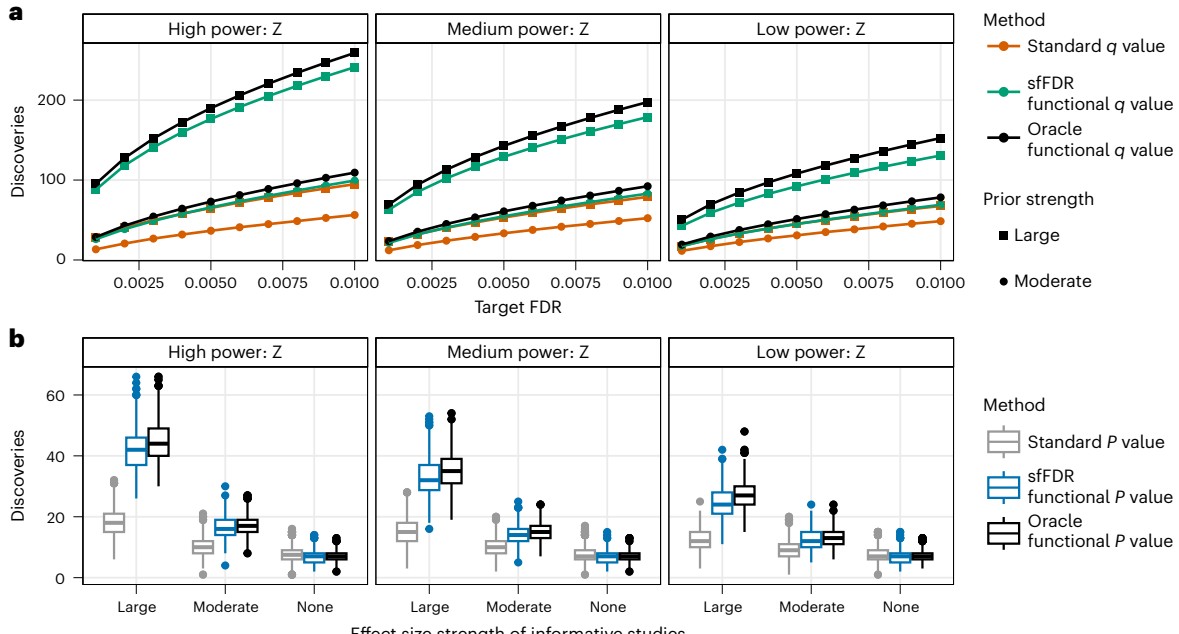

**Fig. 2 | Simulation results for the sfFDR framework in the independent SNP simulation study when the primary study power is 'medium'. a**, The average number of discoveries as a function of the target FDR using the standard $q$ value (dark orange), functional $q$ value from sfFDR (green) and oracle functional $q$ value (black). **b**, The number of discoveries using the standard $P$ value (gray), functional $P$ value from sfFDR (blue) and oracle functional $P$ value (black) at a genome-wide significance threshold of $5 \times 10^{-8}$. We varied the power of the informative studies (columns) and the effect size strength of the informative studies (top plot: shape; bottom plot: $x$ axis). There were a total of 500 replicates at each setting. The box plot shows the median (middle black line), first and third quartiles (box limits), 1.5× interquartile range (IQR; whiskers) and outliers (points).

SNP-level information. We also use the functional local FDR to derive functional local BFs, enabling post-GWAS analyses such as functional fine-mapping to help identify the causal variant in a region (assuming a single causal variant).

## Assessing sfFDR with simulated data

We performed comprehensive simulations to evaluate the sfFDR framework in two settings (Methods). The first setting simulated independent SNPs to allow comparison with other FDR approaches, while the second generated regions of LD to simulate GWAS data. As one of our applications is a rare disease study, we focused on simulating data to reflect the challenging scenario expected in studies of low sample sizes where the genetic signal is sparse.

We simulated the $P$ values for 150,000 independent SNPs in a primary study and three informative studies. The baseline prior probability of a null SNP was 0.98, which reflects our rare disease application (Supplementary Fig. 1). The signal strength (or statistical power) of the studies was varied as 'high', 'medium' and 'low'. As studies of rarer diseases usually have only enough power to identify a small number of non-null SNPs, we assumed that the informative traits influenced the prior probabilities of a small percentage of SNPs. In particular, the informative studies overlapped (shared non-null SNPs with the primary study) with randomly chosen values between 1.25% and 2.50% of the total number of SNPs to represent a sparse overlap with the primary trait. Finally, at the overlapping tests, the informative studies impacted both the prior probability of a SNP being null and the alternative density of the $P$ values with an effect size strength of 'large', 'moderate' and 'none'.

We found that the estimated $q_f$ values controlled the FDR at level 0.01 in all settings (Supplementary Fig. 2), even when the informative traits provided no information on the primary trait. Furthermore, the estimated $q_f$ values had similar power to the true $q_f$ values (referred to as oracle) and substantially improved power compared with the standard $q$ values[18] which were calculated from the qvalue package[22] and do not use the informative studies (Fig. 2a and Supplementary Fig. 3). In general, as the power of the primary or informative study increases, or as the effect sizes in the informative studies become larger, sfFDR leverages more information to increase power. For example, when the power of the informative studies was 'high', the power of the primary study was 'medium' and the effect size strength was 'large', the average number of discoveries from the $q_f$ value was 241 at a target FDR of 0.01. This was much larger than the standard $q$ value (94.5). In the same example, when the power of the informative studies was 'low', the number of discoveries decreased (131) as expected but was still larger than the standard $q$ value (67.6).

We compared the sfFDR framework with other FDR procedures that can incorporate multiple informative variables, namely AdaPT[14], CAMT[16] and an estimator by Boca et al.[13] (referred to as the Boca–Leek method). When evaluating the root mean square error (RMSE) of the estimated $\pi_0(\mathbf{z})$, we found that sfFDR achieves the smallest RMSE across all methods (Supplementary Fig. 4). We then compared estimates of the proportion of truly null hypotheses (the average value of $\pi_0(\mathbf{z})$) and found that CAMT was anti-conservative (predicts more non-null SNPs than exist), AdaPT and Boca–Leek were slightly anti-conservative and sfFDR was conservative (Supplementary Fig. 5). Note that a conservative estimator is more desirable than an anti-conservative estimator as it will not overestimate the amount of signal (which can lead to an inflated FDR).

Because sfFDR relies on a nonparametric surrogate density estimator to handle many informative variables, we compared the performance of the surrogate density estimator with AdaPT and CAMT. While the estimated surrogate density of sfFDR had a smaller RMSE of the log-transformed density values compared with AdaPT and a marginal density estimator, it did not outperform CAMT (Supplementary Fig. 6). However, our data-generating process matched the assumptions behind CAMT's parametric model, and so it is unsurprising that CAMT's parametric density estimator outperformed sfFDR's nonparametric density estimator. In general, a nonparametric density

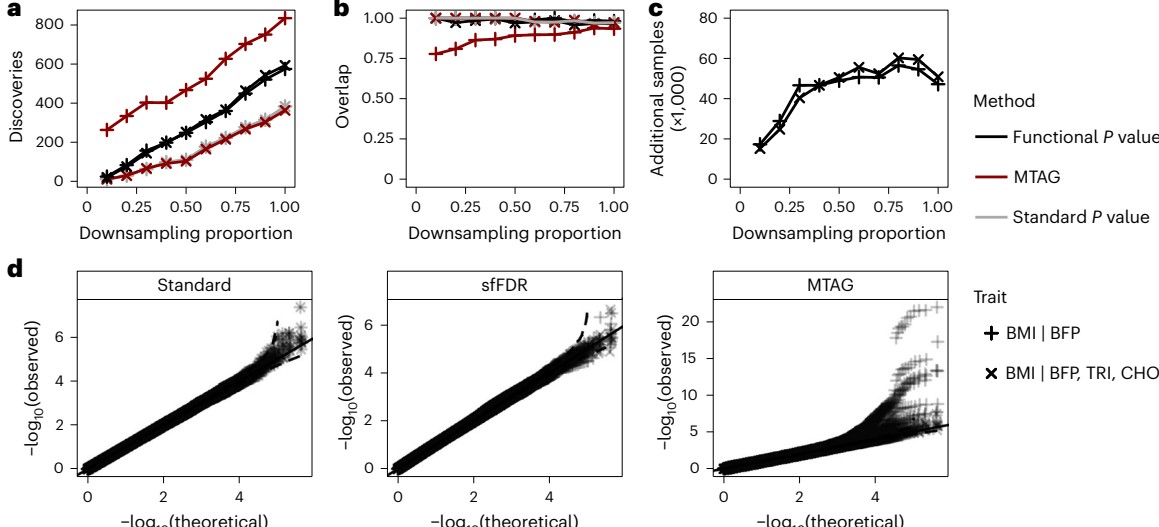

**Fig. 3 | Comparing the functional _P_ value from sfFDR with the _P_ value from MTAG and the standard _P_ value from a GWAS analysis of BMI in the UK Biobank.** The informative traits were BFP and the combination of BFP, triglycerides (TRI) and cholesterol (CHO). We split the UK Biobank data into primary and informative studies, each with a sample size of 190,300. The standard _P_ values were calculated from the primary study (BMI), while the functional _P_ values and MTAG _P_ values also leverage summary statistics of additional obesity-related traits from the informative study. **a**, The number of discoveries as a function of the proportion of the study sample size (downsampling proportion) at a genome-wide significance threshold of $5 \times 10^{-8}$. **b**, The overlap in discoveries (or replication rate) with a meta-analysis approach. **c**, The additional samples required for the GWAS to detect the same number of discoveries as sfFDR. **d**, A quantile–quantile (QQ) plot of the _P_ values from a set of simulated null SNPs after applying the standard analysis, sfFDR and MTAG to the UK Biobank study. For all traits, the standard _P_ values were calculated from a two-sided _t_-test.

estimator offers more flexibility by not assuming an underlying distribution of the data.

When we compared the corresponding estimated $q_f$ values, we found that these methods controlled the FDR (Supplementary Fig. 2), although the FDR was inflated for CAMT when the primary study power was 'low' (Supplementary Fig. 7). Furthermore, sfFDR and CAMT had comparable power and outperformed AdaPT and Boca–Leek across a range of small FDR thresholds (Supplementary Fig. 7). Overall, we found that sfFDR—and, thus, the surrogate density approximation—had improved performance compared with other FDR procedures in our setting.

The estimated $q_f$ value and proportion of truly null tests are then used to construct the $p_f$ value in the sfFDR framework. We found that the estimated $p_f$ value controlled the type I error rate at a significance threshold of $1 \times 10^{-4}$ in the independent SNP simulations (Supplementary Fig. 8). We also evaluated the number of discoveries at a genome-wide significance threshold of $5 \times 10^{-8}$ and compared it with the standard _P_ values (the original _P_ values) and the oracle $p_f$ values (the true $p_f$ values; Fig. 2b and Supplementary Fig. 9). We found that the number of discoveries from the estimated $p_f$ values is close to the oracle $p_f$ values in all settings. As expected, the power improvements from the $p_f$ value compared with the standard _P_ value depend on the primary and informative studies power along with effect size strength. For example, the higher the power of the primary and/or informative studies coupled with a larger effect size strength, the larger the increase in the number of detections from the $p_f$ value.

Finally, we assessed control of the type I error rate and FDR in the dependent SNP setting. We first randomly assigned each independent SNP an LD block size based on the empirical distribution from the UK Biobank (Methods). Given the block size, we then duplicated the _P_ values for the primary and informative studies so that the LD block was perfectly correlated. While this represents an unrealistic scenario, it is a deliberately challenging setting to evaluate estimates in the sfFDR framework because perfect correlations will inflate variability of the type I error rate and FDR compared with real data. Even under such an extreme case, we found that the estimated $p_f$ value and $q_f$ value

from sfFDR controlled the type I error rate and FDR in expectation, respectively (Supplementary Figs. 10–12). As expected, due to the dependence from LD, the variability of the type I error rate and FDR was larger compared with the independent SNP setting. Nevertheless, the estimated $p_f$ value and $q_f$ value had similar variability to the standard _P_ value and _q_ value, respectively.

### Increasing the power in a GWAS of BMI using sfFDR

To investigate the behavior of sfFDR in real data, we split 390,600 unrelated individuals from the UK Biobank into two separate datasets of equal size (Methods): the first (the primary study) was used to detect genetic associations for body mass index (BMI), while the second (the informative study) was used to provide _P_ values for body fat percentage (BFP), triglycerides and cholesterol as informative traits. We downsampled the primary study at different sample sizes to examine the performance of a sfFDR analysis of BMI informed by the three obesity-related traits. We then compared sfFDR with a standard GWAS analysis of BMI and MTAG.

We first evaluated the performance of two SNP selection strategies to train sfFDR, namely selecting SNPs within haplotype blocks or through pruning. For the haplotype block setting, we partitioned the genome into independent (or uncorrelated) blocks and selected a representative SNP either randomly or with the smallest informative trait _P_ value. When comparing the number and replicability of discoveries with a meta-analysis of both datasets (BMI only), we found marginal differences between pruning at multiple correlation thresholds and haplotype block sampling (Supplementary Fig. 13). As our primary application is rare diseases, we selected SNPs on the basis of haplotype blocks with informative trait sampling to maximize coverage of pleiotropic SNPs and minimize computation time (Supplementary Fig. 14).

We then compared sfFDR with an analysis that uses MTAG[21] and a standard GWAS analysis of the primary trait BMI. The number of discoveries from sfFDR was substantially larger than the standard GWAS analysis across a range of sample sizes (Fig. 3a). Furthermore, nearly all of the discoveries made with the $p_f$ values from sfFDR were found by a meta-analysis of both datasets (BMI only; Fig. 3b), suggesting that

the additional discoveries are a subset of those that would be found by increasing the sample size. When applying MTAG to the obesity-related traits, we found that MTAG did not increase the number of discoveries and performed worse than the standard analysis at some sample sizes. As MTAG can have poor performance when applied to many traits[21,23], we considered a simpler scenario where we used only the most informative trait, BFP, in the analysis. In this case, MTAG substantially increased the number of discoveries compared with sfFDR and the standard analysis but with a much lower replication rate (particularly at lower sample sizes; Fig. 3b), implying that a substantial number of discoveries are probably false discoveries. By contrast, the vast majority of discoveries by sfFDR were replicated in the meta-analysis approach. Thus, these results demonstrate the potential of sfFDR to substantially increase the power in GWAS studies by leveraging related traits.

The improvements in statistical power from sfFDR can also be translated in terms of sample size (Fig. 3c). At each downsampling proportion, we predicted the sample size needed for the standard $P$ values to detect the same number of discoveries as the $p_f$ values from sfFDR. The difference in sample sizes between these values is the number of additional samples required for the standard $P$ value to match the discoveries found by the $p_f$ value. We found that the number of additional samples required was quite substantial at each downsampling proportion. For example, at a downsampling proportion of 0.4 (sample size of 76,120), the number of additional samples required was approximately 46,600 (a ~61% increase in sample size). Averaged across all downsampling proportions, we found that the power improvements from sfFDR equated to a ~52% increase in sample size.

We also evaluated the robustness of sfFDR when leveraging summary statistics from other European populations with different LD patterns. In particular, we used the BMI summary statistics from FinnGen[24], GIANT[25] and the Millions Veterans Program (MVP)[26]. When using BMI from the other biobanks as an informative trait, sfFDR and MTAG substantially increased the number of discoveries compared with the standard analysis and nearly all were replicated by a meta-analysis (Supplementary Fig. 15). We found that MTAG outperformed sfFDR at lower samples sizes, while sfFDR outperformed MTAG at larger sample sizes. We note that this is an ideal application of MTAG because the model assumes that the covariance matrix of true SNP effects is the same across all SNPs (that is, the genetic architecture between the primary and informative traits is identical). These results suggest that sfFDR is robust to slight LD differences across the primary and informative traits.

To examine whether using uninformative data leads to false discoveries, we used the obesity-related traits from the UK Biobank study to assess sfFDR in a few 'null' scenarios: (i) the trait values of BMI were permuted and three random chromosomes were replaced with the original BMI summary statistics, (ii) the trait values in the informative study were permuted to generate traits that were uncorrelated with BMI and (iii) the trait values of BMI were permuted to remove any genetic signal. In scenario (i), the $p_f$ values of the null SNPs were well behaved while MTAG $P$ values were substantially inflated when conditioning on BFP (Fig. 3d; see Supplementary Fig. 16 for corresponding $\pi_0(z)$ and $f(p|z)$ estimates). In scenario (ii), the uninformative traits did not systematically inflate the significance of the $p_f$ values or MTAG's $P$ values (Supplementary Figs. 16 and 17). In scenario (iii), there was a substantial inflation of MTAG's $P$ values when conditioning on BFP and a very slight inflation from the $p_f$ values (Supplementary Figs. 16 and 18). Notably, MTAG attempts to stop the software when there is no evidence of a genetic signal in a trait (scenarios (ii) and (iii)) due to the potential of unstable estimates. Even though the slight inflation did not lead to any newly discovered SNPs in sfFDR, we investigated this setting using simulated data and found that the bias arises from overfitting and can be minimized by adjusting the smoothing parameter in the density estimator and setting the proportion of truly null tests as constant (Supplementary Fig. 19). Finally, the observations from

scenarios (i)–(iii) hold regardless of the SNP selection strategy used in sfFDR (Supplementary Fig. 20).

We then evaluated sfFDR in a setting where the conditioning traits were a mixture of uninformative (the permuted null traits from scenario (ii)) and informative (Supplementary Fig. 21). To do so, we fit sfFDR to a mixture of one, two or three informative and uninformative traits. We found that the added uninformative traits did not have a substantial effect on the number of discoveries and nearly all of the discoveries were replicated with the meta-analysis approach. Therefore, including potentially noninformative traits did not result in poor performance in sfFDR.

### Discovering genetic variants in the EGPA study using sfFDR

The sfFDR framework offers potential benefits in the rare disease setting because it is difficult and costly to acquire additional samples to improve power. Therefore, we first examined sfFDR in studies with a small number of cases where independent validation data were available. In particular, we used four diseases in the FinnGen biobank, namely autoimmune thyroiditis (ATH; 688 cases and 424,208 controls), juvenile idiopathic arthritis (JIA; 788 cases and 172,834 controls), myositis (MYO; 932 cases and 357,549 controls) and systemic lupus erythematosus (SLE; 835 cases and 232,612 controls; Methods). In total, the standard analysis identified four lead SNPs (one in ATH and three in SLE) at the genome-wide significance threshold (Supplementary Table 1). Leveraging summary statistics from disease-relevant traits (rheumatoid arthritis[27] and hypothyroidism[28]), sfFDR identified eight additional lead SNPs where the $P$ values of six were genome-wide significant and two were below $5 \times 10^{-7}$ in the independent studies. Thus, we found that the discoveries from sfFDR were replicated in the rare disease setting, in line with our UK Biobank and simulation studies.

Having validated the sfFDR framework in the rare disease setting, we then applied sfFDR to a GWAS of EGPA (676 cases and 6,809 controls)[12], a rare inflammatory disease with a prevalence of around 45.6 per 1,000,000 people in the UK[29]. As most of the accessible patients with EGPA in Europe were used in the study, there were no validation data available. The etiology of EGPA is unknown but is often characterized with other clinical features such as asthma and high eosinophil count[12]. Therefore, these traits are strong candidates to increase power in the EGPA study. We used a publicly available GWAS of childhood-onset asthma (13,962 cases and 300,671 controls)[30], adult-onset asthma (26,582 cases and 300,671 controls)[30] and eosinophil count (172,275 individuals)[31] as our informative studies. After removing nonoverlapping SNPs between EGPA and the informative traits, there were a total of 8,195,277 SNPs used within the sfFDR framework (Methods).

We first evaluated the behavior of the sfFDR framework on EGPA with a set of unrelated traits. Using the permuted null obesity-related traits (unassociated with EGPA) from the UK Biobank analysis, we found that the estimated $p_f$ value from sfFDR tends to be slightly larger than the standard $P$ value (Supplementary Fig. 22). Thus, similar to the above the BMI study, the $p_f$ value from sfFDR conservatively estimates the standard $P$ value for noninformative traits. As the permuted traits do not have any association signal, we also used the original (unpermuted) traits as a set of non-null unrelated traits. On this single realization, the estimated $p_f$ value may be smaller than the standard $P$ value, but on average tends to be slightly larger (Supplementary Fig. 23). Importantly, the estimated $p_f$ values did not discover any newly significant SNPs at the genome-wide significance threshold. Thus, noninformative data do not inflate the type I error rate in the rare disease setting.

We then applied the sfFDR framework to the EGPA study using the EGPA-informative traits (computational time was ~5.40 min on a single core of a Apple M3 processor) and found a substantial increase in the number of discoveries compared with the standard $P$ values (Figs. 4 and 5). We first note that the prior probability of a SNP being null for EGPA varies as a function of the $P$ values of the informative traits, suggesting a shared genetic architecture between traits (Fig. 4a).

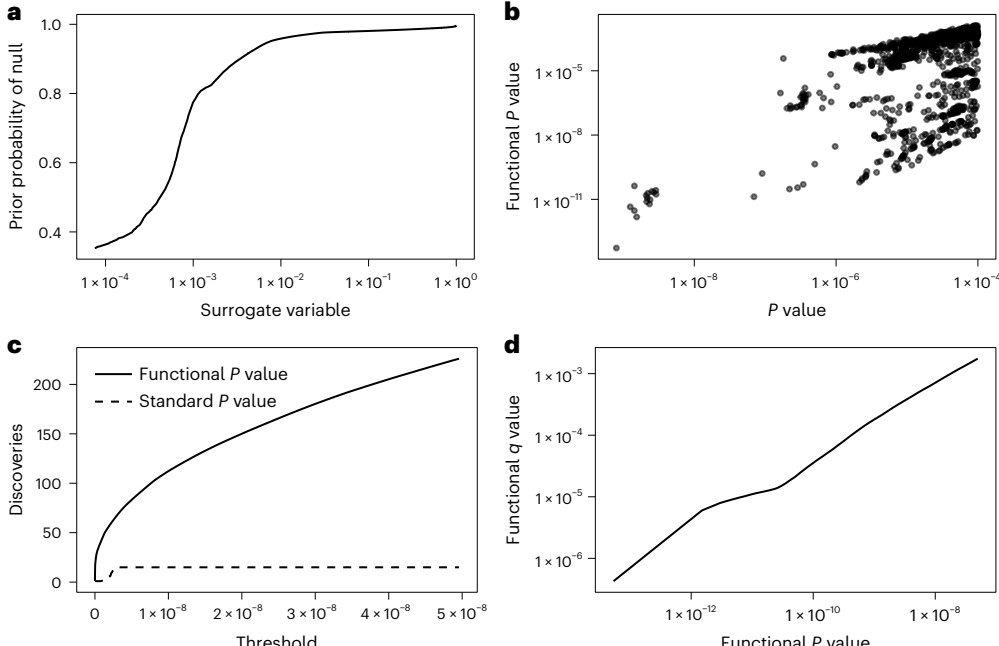

**Fig. 4 | Significance results for the EGPA study under the null hypothesis of no association between genotype and disease. a**, The prior probability of a test being null as a function of the surrogate variable. **b**, The functional *P* value from sfFDR versus the standard *P* value of the study. **c**, The number of discoveries at various *P* value thresholds for the functional and standard *P* values. **d**, The functional *q* value versus functional *P* value relationship. The above plot shows SNPs with standard *P* values below $1 \times 10^{-4}$ in **a** and **b** and functional *P* values below $5 \times 10^{-8}$ in **c** and **d**. The standard *P* values were calculated from a two-sided chi-square test using a linear mixed model (BOLT-LMM).

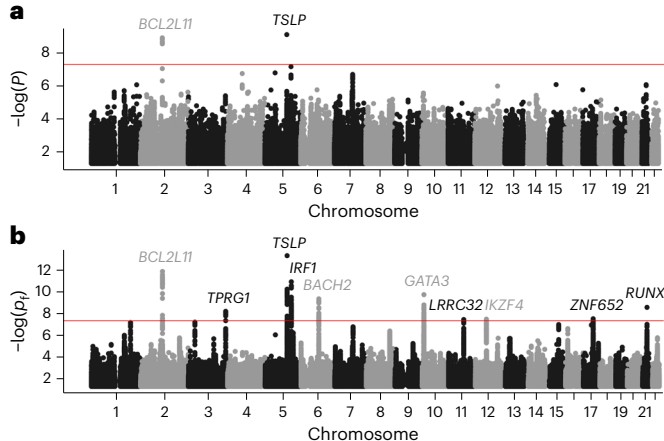

**Fig. 5 | Comparing the functional *P* value from sfFDR with the standard *P* value in the EGPA study. a,b**, Manhattan plot of the standard *P* values (**a**) and the functional *P* values ($p_f$ values) (**b**) from sfFDR. The red line represents the genome-wide significance threshold of $5 \times 10^{-8}$. The lead SNPs were assigned to the nearest genes. Note that *P* values below 0.05 are removed from the plot. The standard *P* values were calculated from a two-sided chi-square test using a linear mixed model (BOLT-LMM).

Furthermore, as a function of significance threshold, the $p_f$ values from sfFDR found substantially more discoveries than the standard *P* values (Fig. 4b,c). For example, at the genome-wide significance threshold, there were 226 discoveries using the $p_f$ values and 15 discoveries using the standard *P* values. Of those discoveries, sfFDR identified ten lead SNPs (or independent associations) instead of two by a standard GWAS analysis (Table 1 and Supplementary Table 2). One feature of the sfFDR framework is that the $p_f$ value can be mapped to the $q_f$ value to control the FDR (Fig. 4d). At the genome-wide significance threshold, we found

that the estimated $q_f$ value is $1.75 \times 10^{-3}$, which implies that there are 0.39 expected false discoveries (defined as a significant SNP that does not tag a causal SNP) in our discovery set of 226 SNPs. Thus, the mapping to a FDR analysis allows the practitioner to choose a data-adaptive significance threshold to control the expected number of false discoveries that they are willing to incur in their analysis.

We focused our analysis on ten lead SNPs with a $p_f$ value below the genome-wide significance threshold (Table 1 and Supplementary Table 2). After assigning SNPs to the nearest gene, we found that the original analysis with the standard *P* values only identified two lead SNPs near *BCL2L1* and *TSLP* while the $p_f$ values from sfFDR identified eight additional genes. At these genes, the lead SNPs were either intergenic (*GATA3*), intronic (*BACH2*, *BCL2L11*, *IRF1*, *RUNX1*, *TPRG1* and *ZNF652*) or upstream (*IKZF4*, *LRRC32* and *TSLP*). Furthermore, the direction of the effect size was consistent across EGPA and the informative traits at these lead SNPs, even though the direction of the effect size is not used by the sfFDR framework.

Many of the new discoveries found by sfFDR are implicated in immune-related processes. For example, *ABI3* (161 kb from rs12952581) and *GATA3* have been linked to eosinophil counts and asthma[32], respectively, as well as *LRRC32,* which encodes the eosinophilic esophagitis-associated TGF-β membrane binding protein *GARP*. In addition, *IRF1* encodes a protein that activates genes involved in pro-inflammatory regulation and has been associated with childhood allergic asthma[33], where it may also have sex-specific effects[34]. Finally, previous work has found that *RUNX1* may be a prognostic marker for some cancers[35,36], and there is evidence that the *RUNX1* transcription factor is involved with Th2 cell differentiation (key for the activation of eosinophils) by decreasing *GATA3* expression[37].

Fine-mapping is a standard post-GWAS analysis to help identify the causal SNP in a region. We fine-mapped each associated region using a standard single causal variant approach with either the functional local BF estimated by sfFDR or the approximate BF[38] (Supplementary Fig. 24 and Methods). Of the two genome-wide

**Table 1 | Functional *P* values (*p*$_f$) and *q* values (*q*$_f$) of the lead SNPs from the EGPA analysis**

| Chr | rsid | Gene | MAF | P | Pleiotropy[a] | p$_f$ | q$_f$ |
|---|---|---|---|---|---|---|---|
| 5 | rs1837253:C>T | *TSLP* | 0.258 | $7.96 \times 10^{-10}$ | [+, +, +] | $5.35 \times 10^{-14}$ | $4.27 \times 10^{-7}$ |
| 2 | rs144569746:T>C | *BCL2L11* | 0.107 | $1.54 \times 10^{-9}$ | [−, −, +] | $1.51 \times 10^{-12}$ | $6.01 \times 10^{-6}$ |
| 5 | rs10066308:A>G | *IRF1* | 0.305 | $6.95 \times 10^{-8}$ | [+, +, +] | $1.33 \times 10^{-11}$ | $1.18 \times 10^{-5}$ |
| 10 | rs7898135:A>C | *GATA3* | 0.283 | $2.72 \times 10^{-6}$ | [+, +, +] | $2.01 \times 10^{-10}$ | $5.74 \times 10^{-5}$ |
| 6 | rs11754356:T>C | *BACH2* | 0.394 | $7.14 \times 10^{-6}$ | [+, +, +] | $4.90 \times 10^{-10}$ | $1.12 \times 10^{-4}$ |
| 21 | rs8133843:A>G | *RUNX1* | 0.373 | $9.69 \times 10^{-7}$ | [−, −, +] | $2.95 \times 10^{-9}$ | $3.51 \times 10^{-4}$ |
| 3 | rs9825301:T>G | *TPRG1* | 0.314 | $4.05 \times 10^{-6}$ | [−, −, +] | $6.82 \times 10^{-9}$ | $5.73 \times 10^{-4}$ |
| 17 | rs12952581:A>G | *ZNF652* | 0.143 | $7.96 \times 10^{-5}$ | [−, +, +] | $3.29 \times 10^{-8}$ | $1.39 \times 10^{-3}$ |
| 12 | rs10876864:A>G | *IKZF4* | 0.416 | $1.19 \times 10^{-4}$ | [+, +, +] | $3.61 \times 10^{-8}$ | $1.47 \times 10^{-3}$ |
| 11 | rs7927997:T>C | *LRRC32* | 0.395 | $2.37 \times 10^{-4}$ | [+, +, +] | $3.86 \times 10^{-8}$ | $1.53 \times 10^{-3}$ |

The informative traits were adult-onset asthma (ASTAO), childhood-onset asthma (ASTCO) and eosinophil count (EOSC). The identifiers for the reference SNP cluster ID (rsid) column are given as rsid:reference_allele>effect_allele. The lead SNP chromosome (Chr) position, nearest gene, minor allele frequency (MAF) and *P* value (*P*) are reported. The standard *P* values were calculated from a two-sided chi-square test using the mixed model association method BOLT-LMM. We adjusted for multiple comparisons using the Bonferroni correction with a genome-wide significance threshold of $5 \times 10^{-8}$. [a]The +/− denotes whether the lead SNP was genome-wide significant/nonsignificant in the informative study (formatted as [ASTAO, ASTCO, EOSC]).

significant regions identified by the standard *P* values, we found 1 and 14 SNPs in the 95% credible set without incorporating informative data compared with 1 and 13 SNPs using sfFDR, respectively. When extended to all the regions found by sfFDR, we found that credible sets were smaller in seven cases (substantially in five cases), unchanged in one and larger in two (Supplementary Table 3). Therefore, a smaller credible set size is not guaranteed. We also calculated the proportion of SNPs in the sfFDR credible sets that overlapped with the credible sets of the informative traits (Supplementary Table 4). Overall, we found that the sfFDR credible sets strongly overlapped with the informative traits (most with eosinophil count) except at the locus in RUNX1 where only 7.70% and 8.10% of the SNPs overlapped with the credible set for adult-onset asthma and childhood-onset asthma, respectively.

## Discussion

The cost of acquiring samples is a limiting factor for discovering genetic variants in GWAS. Our proposed approach, sfFDR, is a cost-effective strategy that leverages pleiotropy to increase power for both small and large studies. Although sfFDR can substantially enhance discovery of genetic variants, FDR approaches have not been widely adopted by the GWAS community despite being commonly used in expression quantitative trait locus mapping. Instead, perhaps due to the abundance of nonreproducible results in earlier candidate gene studies, the preference is to control the FWER in a standard GWAS analysis. Therefore, to help GWAS practitioners leverage the power improvements from fFDR quantities, we derived the functional *P* value, which has a standard *P* value interpretation and can be used in a FWER-controlling procedure while incorporating informative data.

The sfFDR framework allows for a range of significance analyses in a GWAS. More specifically, sfFDR provides estimates of the functional *q* value (a significance measure in terms of the pFDR) and the functional *P* value (a significance measure in terms of the type I error rate). These quantities can be used to map between an FDR threshold and FWER threshold to provide an interpretation for the set of SNPs deemed statistically significant. This is useful for interpreting genetic findings in a GWAS and, more generally, as a data-adaptive way to explore the impact of false discoveries instead of an automatic application of a fixed genome-wide significance threshold. Another FDR quantity estimated by sfFDR, the functional local FDR, provides a simple way to calculate functional local BFs that are key quantities in many post-GWAS analyses. We used it here to perform functional fine-mapping under a single causal variant assumption, but it could also be used to enhance colocalization analysis in the coloc approach[39,40].

Our results have implications for the design of pleiotropy-informed significance analyses. As expected, practitioners should use informative traits with large sample sizes to increase the power of the primary study. Fortunately, there is a large collection of GWAS summary statistics in publicly available repositories for thousands of complex traits[3,41], although selecting the informative traits a priori will require careful consideration to avoid model selection (and fitting) problems. Identifying polygenic traits that have a similar genetic architecture with the primary trait will provide the largest increase in power, and so traits that are too dissimilar (or with little polygenicity) will not be ideal informative traits. In our rare disease analysis, we chose informative traits based on known co-morbidities among the patients. Another strategy is to select informative traits based on the estimated genetic correlation with the primary trait. However, we caution against a 'shotgun' approach where the largest correlations among hundreds of traits are chosen as this can lead to selection bias. Importantly, our results showed that sfFDR is robust to the inclusion of noninformative traits. Thus, sfFDR can integrate multiple sets of candidate informative traits with different genetic architectures. By contrast, MTAG did not improve upon the standard analysis when conditioning on the three obesity-related traits and can also inflate the type I error rate. While our method can incorporate many informative traits, handling a very large number of informative traits may require dimensionality reduction (for example, principal component analysis[42] or sliced inverse regression[43]), variable selection or regularization to reduce the computational time and allow for stable model fitting in sfFDR (Supplementary Fig. 14).

Our approach is not immune to sources that may bias summary statistics such as ancestry[44] or nonrandom sampling with respect to the reference population (for example, participation bias[45]). In particular, if ancestry is unaccounted for in both the primary and informative studies, then there is a risk that ancestry-informative SNPs could be elevated by sfFDR. Therefore, it is important to consider only studies that adopt robust analytical strategies. Furthermore, while our results suggest that sfFDR is robust to slight LD differences across the primary and informative traits, we do not recommend using summary statistics from populations that are very different (in terms of ancestry) than the primary study.

There are a few important observations when applying the sfFDR framework to GWAS data. First, as our primary application is rare diseases, we trained sfFDR using haplotype blocks with informative trait sampling to maximize coverage of pleiotropic SNPs. However, for more common diseases, our results suggest there is flexibility in how to choose SNPs (for example, using high pruning correlation thresholds or potentially using all of the SNPs), and in such cases, the

primary limitation is computational time (Supplementary Fig. 14). Second, while we found the surrogate variable based on the functional proportion of truly null hypotheses performed well in this study, it is possible that there may be better surrogate variable choices or the nonparametric density estimation could be extended to incorporate multiple variables[15]. Third, as it is not possible to distinguish whether a (tagged) SNP is a true discovery or is capturing a nearby causal SNP due to LD, we defined a true discovery as a SNP that either tags or is the causal SNP. Our results showed that the functional $P$ value controls the type I error rate when either the causal variant is measured (as in the simulation study) or tagged (as in the UK Biobank analysis). Finally, we have assumed that subjects in the primary study are not also included in the informative studies, so that the sets of $P$ values are independent under the null hypothesis.

While our emphasis is on leveraging pleiotropy from GWAS summary statistics, there is a large body of existing datasets to further increase statistical power, such as functional annotations in various cell types or states, expression-level data or minor allele frequency. As such, we anticipate that sfFDR will have broader applications in genome-wide studies as a general framework that integrates informative data and provides a cost-effective way to improve power.

## Methods

### Overview
We first review the theory behind the fFDR framework[15] and then introduce the functional $P$ value. Consider a GWAS study with $P$ values $P_i$ for $i = 1, 2, …, m$ SNPs. We initially assume that the $P$ values are approximately independent (via pruning or clumping) and identically distributed random variables (LD considered in 'Extending the sfFDR framework to include SNPs in LD' section). The $P$ values follow a two-group mixture model composed of SNPs that are not associated (the null) with probability $\pi_0$ or are associated (the non-null or alternative) with probability $1 - \pi_0$. Let the status of a SNP that is null be denoted by $H_i = 0$ and one that is non-null be denoted by $H_i = 1$. Suppose that there are $d$ sets of informative GWAS summary statistics, $\mathbf{Z}_i = (Z_{i1}, Z_{i2}, …, Z_{id})$, that can influence (1) the prior probability of a SNP being null, $(H|\mathbf{Z} = \mathbf{z}) \sim$ Bernoulli$(1 - \pi_0(\mathbf{z}))$, and/or (2) the distribution of the $P$ values under the alternative hypothesis, $(P|H = 1, \mathbf{Z} = \mathbf{z}) \sim F_1(\cdot|\mathbf{z})$, where $F_1$ is some distribution stochastically smaller than the uniform distribution. As we assume that individuals from the primary study are not in the informative studies, the summary statistics do not impact the $P$ values under the null hypothesis, $(P|H = 0, \mathbf{Z} = \mathbf{z}) = (P|H = 0) \sim$ Uniform$(0, 1)$. It is worth noting that the informative studies can share individuals between themselves.

Given the above assumptions, we can define a decision rule that incorporates the $P$ values and summary statistics to identify statistically significant SNPs. In particular, without loss of generality, we assume that the informative statistics are transformed to be uniformly distributed on the unit interval by using ranks. The significance region, $\Gamma \in [0,1]^{1+d}$, for the statistic $T = (P, \mathbf{Z})$ is defined as

$$\Gamma_\tau = \left\{ (p, \mathbf{z}) \in [0,1]^{1+d} : \Lambda(p, \mathbf{z}) \leq \tau \right\}, \tag{1}$$

where $\tau \in [0,1]$ is a significance threshold and

$$\Lambda(p, \mathbf{z}) = \Pr(H = 0|T = (p, \mathbf{z}))$$
$$= \frac{f(p|H=0,\mathbf{z}) \Pr(H=0|\mathbf{z})}{f(p|\mathbf{z})} = \frac{\Pr(H=0|\mathbf{z})}{f(p|\mathbf{z})} = \frac{\pi_0(\mathbf{z})}{f(p|\mathbf{z})} \tag{2}$$

is the probability that a SNP is a false discovery given the observed data (the posterior error probability). Intuitively, the significance region classifies a set of SNPs with posterior error probabilities less than or equal to some threshold $\tau$ as statistically significant. The posterior error probability in this context is referred to as the functional local FDR, and it is the optimal statistic for the Bayes rule with Bayes error[15].

As such, our strategy to optimally incorporate the summary statistic data is based on the functional local FDR.

Using the significance region in equation (1), we can construct the functional $q$ value ($q_f$ value) and $P$ value ($p_f$ value), which are different measures of significance for a SNP. Formally, the $q_f$ value is the minimum positive FDR (pFDR; closely related quantity to FDR) incurred when calling a SNP statistically significant[15,18] while the $p_f$ value is the minimum type I error rate incurred when calling a SNP statistically significant. We note that these quantities have a Bayesian interpretation: the pFDR is the probability of a SNP being null given that it is classified as statistically significant[20], pFDR$(\Gamma_\tau) = \Pr(H = 0|T \in \Gamma_\tau)$, and the type I error rate is the probability of a SNP being classified as statistically significant given that it is null, $\Pr(T \in \Gamma_\tau|H = 0)$. Thus, for an observed statistic $t = (p, \mathbf{z})$, we can express the $q_f$ value as

$$q_f(p, \mathbf{z}) = \inf_{\{\Gamma_\tau : t \in \Gamma_\tau\}} \text{pFDR}(\Gamma_\tau) = \text{pFDR}(\Gamma_{\Lambda(p,\mathbf{z})}), \tag{3}$$

and the $p_f$ value as

$$p_f(p, \mathbf{z}) = \inf_{\{\Gamma_\tau : t \in \Gamma_\tau\}} \Pr(T \in \Gamma_\tau|H = 0) = \Pr(T \in \Gamma_{\Lambda(p,\mathbf{z})}) \times \frac{q_f(p, \mathbf{z})}{\pi_0}, \tag{4}$$

where $\Pr(T \in \Gamma_{\Lambda(p,\mathbf{z})})$ is the cumulative distribution function. While the definition of the $p_f$ value is the same as a standard $P$ value, we call it 'functional' to emphasize that it is a function of the informative data.

The $q_f$ value and $p_f$ value are complementary quantities in a significance analysis: the former allows a researcher to decide the expected number of false discoveries they are willing to incur in the study while the latter allows for a standard $P$ value interpretation. We can use such measures of significance to identify statistically significant SNPs by either rejecting SNPs with a $p_f$ value below a genome-wide significance threshold or a $q_f$ value below a desired FDR level. The mapping between the $q_f$ value and $p_f$ value provides different interpretations for the set of statistically significant SNPs and thus connects a standard GWAS analysis to a FDR analysis while incorporating the informative data. In the next section, we discuss how to construct estimates of the functional local FDR, $q_f$ value and $p_f$ value.

### Estimating the functional local FDR, $q_f$ value and $p_f$ value in the sfFDR framework
We first review construction of the $q_f$ value and $p_f$ value and then estimation in the sfFDR framework. Given the significance region defined by equation (1), the $q_f$ value for the $i$th SNP is

$$q_f(p_i, \mathbf{z}_i) = \frac{1}{|\mathcal{S}_i|} \sum_{j \in \mathcal{S}_i} \Lambda(p_j, \mathbf{z}_j), \tag{5}$$

where $\mathcal{S}_i = \{ j : \Lambda(p_j, \mathbf{z}_j) \leq \Lambda(p_i, \mathbf{z}_i) \}$ is the set of SNPs with functional local FDRs less than or equal to the value of the $i$th SNP[15]. The corresponding $p_f$ value is then

$$p_f(p_i, \mathbf{z}_i) = \Pr\left( T \in \Gamma_{\Lambda(p_i, \mathbf{z}_i)} \right) \times \frac{q_f(p_i, \mathbf{z}_i)}{\pi_0}. \tag{6}$$

As the $q_f$ value and $p_f$ value can be constructed from the functional local FDR, the primary quantities to estimate are $\pi_0(\mathbf{z})$ and $f(p|\mathbf{z})$.

The sfFDR framework provides estimates of the above quantities by extending the functional FDR framework to incorporate multiple GWAS summary statistics. In particular, we estimate $\pi_0(\mathbf{z})$ by minimizing the mean integrated squared error using a GAM and $f(p|\mathbf{z})$ nonparametrically using a local likelihood kernel density estimator (KDE). We describe further details below and extend our discussion to include LD in the 'Extending the sfFDR framework to include SNPs in LD' section.

**Estimation of $\pi_0(\mathbf{z})$.** We extend the GAM method from Chen et al.[15] to multiple informative variables. Let $\eta_\lambda(\mathbf{z}) = \mathbf{1}_{\{P>\lambda|\mathbf{Z}=\mathbf{z}\}}$ denote a binary response variable where it follows that $E[\eta_\lambda(\mathbf{z})] = \Pr(P > \lambda|\mathbf{Z} = \mathbf{z}) \geq \Pr(P > \lambda|H = 0, \mathbf{Z} = \mathbf{z})\Pr(H = 0|\mathbf{Z} = \mathbf{z}) = (1-\lambda)\pi_0(\mathbf{z})$ for some $\lambda \in [0,1)$. Given a set of informative variables, the general model is

$$\text{logit}(E[\eta_\lambda(\mathbf{z})]) = \beta_0 + \sum_{k=1}^{d} f_k(z_k), \tag{7}$$

where $\text{logit}(x) = \log(\frac{x}{1-x})$, $\beta_0$ is a constant and $f_k(z_k)$ is some function of the $k$th informative variable. In this work, we use a natural cubic spline with knots chosen at specified quantiles (described below). Note that the above model allows for nonlinear relationships and conservatively estimates the prior probabilities (or functional proportion of truly null hypotheses) at a given $\lambda$, that is, $\frac{E[\eta_\lambda(\mathbf{z})]}{1-\lambda} \geq \pi_0(\mathbf{z})$.

We implement the following algorithm to estimate the functional proportion of truly null hypotheses. We first place the knots at small values (or lower quantiles) of $z_k$, which are regions that are likely to contain alternative $P$ values. These regions will vary based on the signal density and power of the informative studies. We then fit the above model at $\lambda = 0.05, 0.1, \ldots, 0.9$ and choose the fit that minimizes the mean integrated squared error[15]. The estimated functional proportion of truly null hypotheses at this minimum $\lambda$ is $\hat{\pi}_0(\mathbf{z}; \lambda_{\min}) = \frac{\hat{E}[\eta_{\lambda_{\min}}(\mathbf{z})]}{1-\lambda_{\min}}$, and, as discussed above, is conservative. We note that if the test statistics are used (instead of $P$ values), then the knots should be placed where the signal is expected (the lower and/or upper tails of the distribution).

**Estimation of $f(p|\mathbf{z})$.** A challenge with nonparametric density estimation is that the joint density is difficult to estimate as the number of variables increases. We circumvent this difficulty by constructing a surrogate (or compressed) variable to reduce the dimensionality. In particular, we construct a surrogate variable based on $\pi_0(\mathbf{z})$: let $r_i = r_i^*/m$ be the uniform quantile transformation of $\pi_0(\mathbf{z}_i)$ for $i = 1, 2, \ldots, m$, where $r_i^*$ is the rank of the $i$th hypothesis (any ties are randomly assigned). We then estimate the density of $f(p|r) = \frac{f(p,r)}{f(r)} = f(p,r)$ instead of $f(p|\mathbf{z})$, which is more tractable when there are many informative variables. To estimate $f(p, r)$, we use a local likelihood KDE on the probit-transformed scale[15,46]. The nearest-neighbor smoothing parameter is chosen to be the estimated proportion of truly alternative tests of the $P$ values, and so the smoothing neighborhood covers $100 \times (1 - \pi_0)\%$ of the data. Note that if $1 - \pi_0 < 0.02$, then we set the smoothing parameter to be 0.02.

In summary, we approximate the functional local FDR as

$$\Lambda(p, \mathbf{z}) \approx \pi_0(\mathbf{z}) \frac{f(r)}{f(p,r)} = \frac{\pi_0(\mathbf{z})}{f(p,r)}, \tag{8}$$

where the surrogate variable $r$ is uniform quantile transformation of $\pi_0(\mathbf{z})$ and $f(r) = 1$. We refer to the above approximation as surrogate functional FDR (sfFDR) to emphasize that it is based on the surrogate variable $r$. Importantly, sfFDR reduces the dimensionality for tractable nonparametric density estimation. With the estimated (approximate) functional local FDR, we can then estimate the $q_f$ value and $p_f$ value as $\hat{q}_f(p_i, \mathbf{z}_i) = \frac{1}{|\mathcal{S}_i|}\sum_{j \in \mathcal{S}_i} \hat{\Lambda}(p_j, \mathbf{z}_j)$ and $\hat{p}_f(p_i, \mathbf{z}_i) = \hat{\Pr}\left(T \in \Gamma_{\hat{\Lambda}(p_i, \mathbf{z}_i)}\right) \times \frac{\hat{q}(p_i, \mathbf{z}_i)}{\hat{\pi}_0}$, respectively. We note that $\hat{\Pr}\left(T \in \Gamma_{\hat{\Lambda}(p_i, \mathbf{z}_i)}\right)$ is the empirical CDF of the functional local FDRs and that the prior probability can be estimated as $\hat{\pi}_0 = \frac{1}{m}\sum_{i=1}^{m} \hat{\pi}_0(\mathbf{z}_i; \lambda_{\min})$ or using the maximum $q_f$ value in the study.

### Extending the sfFDR framework to include SNPs in LD

Thus far, we have assumed that a subset of SNPs have been selected to be approximately independent via pruning or clumping. While this may be useful to understand a region's contribution to phenotypic variation, it is difficult to select the 'best' representative SNP in an LD region. Therefore, we extend the sfFDR framework to circumvent such

difficulty by providing a measure of significance for each SNP (including SNPs in LD) while incorporating the informative data.

To extend the sfFDR framework, we first model the proportion of truly null hypotheses, $\pi_0(\mathbf{z})$, and the joint density, $f(p, r)$, on a set of LD-independent SNPs and then use the fitted curves to predict the corresponding values of the left-out SNPs in LD (Fig. 1). More specifically, we identify a subset of LD-independent SNPs via pruning or clumping, or by using the informative traits (see 'Application to EGPA study' section). Using the LD-independent SNPs, we apply the GAM method to estimate $\pi_0(\mathbf{z})$ and use the fitted curve to predict $\pi_0(\mathbf{z})$ of the left-out SNPs. After constructing the surrogate variable from the estimated $\pi_0(\mathbf{z})$, the joint density, $f(p,r)$ and the marginal density, $f(r)$, are estimated using the LD-independent SNPs. We note that the marginal density of the surrogate variable may not follow a uniform distribution when including SNPs in LD. As such, we estimate the marginal density using a nonparametric KDE. Finally, the density values for the left-out SNPs are predicted from these fitted density curves. We can then estimate the functional local FDR ($\Lambda(p, \mathbf{z}) \approx \pi_0(\mathbf{z}) \frac{f(r)}{f(p,r)}$) along with the corresponding $q_f$ value and $p_f$ value as outlined in the 'Estimating the functional local FDR, $q_f$ value and $p_f$ value in the sfFDR framework' section.

### Fine-mapping with the functional local FDR

The FDR quantities estimated from the sfFDR framework can be used to perform fine-mapping under the assumption that there is a single causal variant in a region[47]. More specifically, suppose there are $j = 1, 2, \ldots, L$ variants in a region of interest. The functional local BF can be expressed in terms of the functional local FDR as

$$\begin{aligned}\text{BF}(p, \mathbf{z}) &= \frac{\Pr(H=0)}{\Pr(H=1)} \times \frac{\Pr(H=1|p,\mathbf{z})}{\Pr(H=0|p,\mathbf{z})}, \\ &= \frac{\pi_0}{1-\pi_0} \times \frac{1-\Lambda(p,\mathbf{z})}{\Lambda(p,\mathbf{z})},\end{aligned} \tag{9}$$

where $\pi_0$ is the prior probability of the null hypothesis and $\Lambda(p, \mathbf{z})$ is the functional local FDR. Under the assumption of a single causal variant in the region, the posterior probability (PP) for the $i$th SNP is

$$\text{PP}(p_i, \mathbf{z}_i) = \frac{\text{BF}(p_i, \mathbf{z}_i)}{\sum_{j=1}^{L} \text{BF}(p_j, \mathbf{z}_j)}, \tag{10}$$

where we have implicitly assumed a uniform prior on any variant being the causal variant[47]. Therefore, the sfFDR framework provides estimates of the functional local BF and the corresponding PP for each SNP to help identify the causal locus. More generally, because the sfFDR framework incorporates SNP-level data, it is also a framework to perform functional fine mapping. Note that the functional local BFs can also be used in any post-GWAS analysis in place of approximate BFs[38]. For example, while we do not explore it in this work, the functional local BFs estimated by sfFDR can also be used to perform colocalization[39,40] while integrating informative data.

### Simulation study

We conducted comprehensive simulations to assess the performance of the sfFDR framework. We simulated 150,000 independent hypotheses for the primary study with corresponding summary statistics for $k = 1, 2, 3$ informative studies. The proportion of null hypotheses was simulated as $\pi_0^{(k)} \sim \text{Uniform}[1 - \gamma, 1 - \gamma/2]$, where the first $1 - \pi_0^{(k)}$ $P$ values were generated from the alternative distribution and the remaining were generated from the null distribution (standard uniform distribution). We fixed the number of shared tests from the alternative hypothesis (or overlap) between the informative studies and our primary study to be $\gamma = 0.025$ (a low level of overlap). Under the alternative hypothesis, we assumed the $P$ values followed a Beta$(\alpha, 5)$, where $\alpha = 2$ for the 'high' signal strength (or density), $\alpha = 3$ for the 'medium' signal strength and $\alpha = 4$ for the 'low' signal strength cases. We describe below how the

informative studies $P$ values (denoted by $\mathbf{z} = (z_1, z_2, z_3)$) influenced the prior probability $\pi_0(\mathbf{z})$ and the alternative density $f_1(p|\mathbf{z})$ of the primary study $P$ values.

**Prior probability $\pi_0(\mathbf{z})$.** The relationship between the probability of a hypothesis test being truly null and the informative summary statistics was generated as follows. Define the function

$$\phi^{(k)}(z_k) = \begin{cases} 0.98 \times \left(\frac{z_k}{\pi_0^{(k)}}\right)^a, & \text{if } z_k < \pi_0^{(k)} \text{ and } H^{(k)} = 1 \\ 0.98, & \text{otherwise} \end{cases}$$

where $a = 0.6$ for the 'large' effect size strength case, $a = 0.3$ for the 'moderate' effect size strength case and $H^{(k)} = 1$ for a test that is truly alternative in the $k$th informative study. The average of these components are then used to construct the prior probability of a hypothesis being null,

$$\pi_0(\mathbf{z}) = \frac{\sum_{k=1}^{3} \phi^{(k)}(z_k)}{3}.$$

This relationship reflects the expected behavior where the prior probability decreases as the informative $P$ value decreases for shared alternatives. Using the prior probabilities, we then drew the true status of the $i = 1, 2, \ldots, m$ hypotheses as $(H_i|\mathbf{Z}_i = \mathbf{z}_i) \sim \text{Bernoulli}(1 - \pi_0(\mathbf{z}_i))$. Under the null hypothesis, the $P$ values were generated from a standard uniform distribution, $(P|H = 0, \mathbf{Z} = \mathbf{z}) \sim \text{Uniform}(0, 1)$. We describe the distribution under the alternative hypothesis below.

**Alternative density $f_1(p|\mathbf{z})$.** Under the alternative hypothesis, the distribution of the $P$ values varied as a function of the informative variables. In particular, define the function

$$\omega^{(k)}(z_k) = \begin{cases} \frac{z_k}{\pi_0^{(k)}}, & \text{if } z_k < \pi_0^{(k)} \text{ and } H^{(k)} = 1 \\ 1, & \text{otherwise} \end{cases}$$

and

$$\omega^*(\mathbf{z}) = \frac{\sum_{k=1}^{3} \omega^{(k)}(z_k)}{3}.$$

We assumed that the $P$ values follow a beta distribution under the alternative hypothesis, $(P|H = 1, \mathbf{Z} = \mathbf{z}) \sim \text{Beta}(\alpha(\mathbf{z}), 5)$, where $\alpha(\mathbf{z}) = \alpha_0 - c \times (1 - \omega^*(\mathbf{z}))$. The parameter $\alpha_0$ controls the signal strength (or density) of the alternative distribution and the parameter $c$ controls the effect size strength of the informative summary statistics. We considered $\alpha_0 = 0.3$ for the 'high' signal strength, $\alpha_0 = 0.4$ for the 'medium' signal strength and $\alpha_0 = 0.5$ for the 'low' signal strength cases. We set the parameter $c$ equal to $\alpha_0/2$ when the informative studies had a 'large' effect size strength and $\alpha_0/4$ when the informative studies had a 'moderate' effect size strength.

In total, there were 500 replicates at each combination of primary study signal strength, informative study signal strength and the effect size strength of the informative studies. We also considered the scenario where the informative summary statistics have no impact on the primary $P$ values and so $\pi_0(\mathbf{z}) = 0.98$ and $\alpha = \alpha_0$. For the $p_f$ values, we evaluated the type I error rate at a threshold of $1 \times 10^{-4}$ and the power at $5 \times 10^{-8}$. For the $q_f$ values, we evaluated the FDR at level 0.01 and the accuracy of the estimated proportion of truly null tests. We then compared the sfFDR framework with three different FDR procedures that can incorporate informative variables, namely AdaPT[14], CAMT[16] and an estimator by Boca et al. (2018; referred to as the Boca–Leek method)[13]. The default settings of each software were used where the inputs were standardized across implementations. To assess our method under

LD, for the $i = 1, 2, \ldots, m$ independent tests, we replicated the $P$ value and corresponding informative summary statistics $s_i$ times, where $s_i$ is drawn from the empirical distribution of the LD block sizes estimated using the UK Biobank (see 'UK Biobank study' section). This reflects an extreme scenario where the SNPs in LD are perfectly correlated.

## UK Biobank study

The UK Biobank is a repository of genetic, lifestyle and health information for over half a million UK participants[48,49]. Our analysis used four obesity-related traits that were rank-based inverse normal transformed, namely BMI, BFP, cholesterol and triglycerides. We restricted our analysis to 380,600 unrelated individuals with British ancestry. We then split the UK Biobank into two equal parts of size 190,300, where one part was used as the 'primary' study and the other was the 'informative' study.

Our trait of interest in the primary study was BMI and the informative traits were BFP, cholesterol and triglycerides. We downsampled the primary study to 10%, 20%, ..., 90%, 100% of the original sample size to study the impact of lower statistical power in our procedure. We applied the following processing steps to all downsampled datasets. Using the genotyped data (autosomes only), SNPs were filtered in PLINK with a MAF < 0.001, Hardy–Weinberg equilibrium $P$ value threshold of $<1 \times 10^{-10}$ and a genotype missingness rate >0.05. Because we compared sfFDR with MTAG (with default settings), we focused on SNPs outside the MHC (the region between 24 Mb and 45 Mb) to avoid any issues with large effect sizes impacting MTAG. We then applied PLINK[50] for association testing while adjusting for sex, age and the top 20 principal components provided by the UK Biobank to account for ancestry.

There were a few different 'null' simulation settings considered. We implemented a scenario where the informative traits were permuted to be uncorrelated with BMI. In total, there were ten permuted null datasets analyzed. We then considered another setting where we permuted BMI (ten times) and replaced three randomly selected chromosomes with the original BMI summary statistics. We designed the simulation this way so that we knew which SNPs were non-null while maintaining the genetic correlation across traits at the three selected chromosomes. Finally, we used the permuted null BMI summary statistics and the original (non-null) summary statistics of the informative traits. In this case, there is no genetic signal in the BMI summary statistics, and so the proportion of truly null tests is 1.

There were two types of SNP selection strategy considered in this work: using haplotype blocks and pruning. For the haplotype block setting, LD-independent SNPs were determined by using predefined haplotype blocks constructed using the linkage-disequilibrium adjusted kinship (LDAK) method[51]. More specifically, within each haplotype block, we performed hierarchical clustering using a random subset of 40,000 individuals from the UK Biobank to identify clusters of uncorrelated SNPs. In total, there were 161,207 'independent' clusters at a pruned correlation threshold of 0.01. At each cluster, a representative SNP was selected either randomly or based on having the smallest informative trait $P$ value (see 'Application to EGPA study' section). In the pruned SNP setting, we applied PLINK varying the squared correlation threshold to 0.05, 0.1, 0.2, 0.3 and 0.4 (calculated on a sliding window of the 1,000 closest SNPs). This resulted in training sfFDR on 113,834, 180,915, 272,351, 336,932 and 380,552 SNPs. In addition, we also used all of the SNPs to train sfFDR. Finally, in our implementation of the GAM model, we fit a natural cubic spline to the informative trait $P$ values with knots placed at the 0.005, 0.025, 0.01, 0.05 and 0.1 quantiles.

## Validating sfFDR in the rare disease setting

The limiting factor in the rare disease setting is the cost of acquiring samples. As such, the number of collected cases is often much lower than studies of more common diseases. To examine sfFDR in a similar setting, we analyzed four diseases with a low number of cases in the FinnGen biobank for which independent validation data was available[24]: ATH (688 cases and 424,208 controls), JIA (788 cases and 172,834

controls), MYO (932 cases and 357,549 controls) and SLE (835 cases and 232,612 controls). We used the summary statistics from release versions R5 (JIA), R7 (SLE) and R12 (MYO and ATH) and leveraged the summary statistics of rheumatoid arthritis (14,361 cases and 43,923 controls)[27] and hypothyroidism (405,357 individuals)[28] as clinically relevant traits. After filtering SNPs with MAF <0.01, there were 2,970,059, 2,911,726, 2,970,147 and 2,914,017 SNPs used in the analysis for ATH, JIA, MYO and SLE, respectively.

### Application to EGPA study

We applied the sfFDR framework to a GWAS of EGPA. To illustrate our method on this rare disease, we used the *P* values from a publicly available GWAS with 676 cases and 6,809 controls (see Lyons et al.[12] for analysis details). Because the EGPA study provided only discrete *P* values (two significant digits), we recalculated the *P* values using the publicly available effect sizes and standard errors and found a strong concordance with the published *P* values (Supplementary Fig. 25). In total, there were 9,246,221 typed or imputed autosomal variants with INFO scores greater than 0.8 included in the analysis. The informative GWAS summary statistics were from clinically relevant features of EGPA, namely childhood-onset asthma (13,962 cases and 300,671 controls)[30], adult-onset asthma (26,582 cases and 300,671 controls)[30] and eosinophil count (172,275 individuals)[31]. See the referenced publications for additional information on quality-control steps. After removing SNPs in the MHC region and nonoverlapping SNPs between EGPA and the informative traits, there were a total of 8,195,277 SNPs used in our analysis. Note that all of the studies calculated *P* values from a two-sided test using the mixed model association method BOLT-LMM[52].

In our analysis, we considered 161,207 'independent' regions of the genome that were identified by a hierarchical clustering algorithm described in the 'UK Biobank study' section. To potentially increase the coverage of pleiotropic SNPs, we selected LD-independent SNPs as follows. For each region, if any of the informative traits *P* values were below 0.001, then we selected the SNP that had the smallest *P* value among the informative traits. Otherwise, we randomly selected a SNP in the region. When modeling the proportion of truly null hypotheses, we fit a natural cubic spline to the informative traits *P* values with knots placed at the 0.005, 0.025, 0.01, 0.05 and 0.1 quantiles. Finally, we used the original and permuted obesity-related traits (BFP, cholesterol and triglycerides) from the UK Biobank (see 'UK Biobank study' section) to assess whether sfFDR recovers the original *P* values when the informative traits are unrelated to EGPA.

### Ethics statement

The research in this work abides by the Helsinki Declaration. The North West Multi-centre Research Ethics Committee (MREC) approved the protocols of the UK Biobank study. The REC reference number is 06/MRE08/65. The FinnGen Biobank collected informed consent for research via the Finnish Biobank Act. The Central VA Institutional Review Board provided appropriate consent and protocol approvals for the MVP Biobank.

### Reporting summary

Further information on research design is available in the Nature Portfolio Reporting Summary linked to this article.

### Data availability

The asthma (GCST007800, GCST007799), eosinophil count (GCST004606), rheumatoid arthritis (GCST002318), hypothyroidism (GCST90013893) and EGPA (GCST009250) GWAS summary statistics are publicly available to download at https://www.ebi.ac.uk/gwas. The MYO, SLE, JIA and ATH GWAS summary statistics are publicly available to download at https://finngen.gitbook.io/data-download. The Million Veteran Program summary statistics can be downloaded through dbGAP at https://dbgap.ncbi.nlm.nih.gov/ (accession number phs002453). This research has been conducted using the UK Biobank Resource under Applied Number 98032. Access to the UK Biobank data can be requested at https://www.ukbiobank.ac.uk/enable-your-research/apply-for-access. Source data are provided with this paper. Source data for Figs. 2–5 are available via Zenodo at https://doi.org/10.5281/zenodo.15754143 (ref. 53).

### Code availability

sfFDR is publicly available in the R package sffdr and can be downloaded at https://github.com/ajbass/sffdr (most recent version)[54] or https://cran.r-project.org/web/packages/sffdr/index.html (release). The code to reproduce the results in this work is available at https://github.com/ajbass/sffdr_manuscript (ref. 55).

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

## Acknowledgements

This work was supported by the Wellcome Trust grants WT220788 (C.W.) and WT219506 (C.W. and A.J.B.) and the MRC grants MC_UU_00002/4 (C.W.) and MC_UU_00040/01 (C.W.). The funders had no role in the study design, data collection and analysis, decision to publish or preparation of the manuscript. We also acknowledge the participants and investigators of the FinnGen, MVP and UK Biobank studies.

## Author contributions

A.J.B. and C.W. designed the study and developed the methodology. A.J.B. performed the analyses and developed the software. C.W. conceptualized and supervised the study. All authors read and approved the final manuscript.

## Competing interests

C.W. has received funding from GSK and MSD and is a part-time employee of GSK. These companies had no input into this work. A.J.B. declares no competing interests.

## Additional information

**Correspondence and requests for materials** should be addressed to Andrew J. Bass or Chris Wallace.

Chris Wallace

# Reporting Summary

## Statistics

For all statistical analyses, confirm that the following items are present in the figure legend, table legend, main text, or Methods section.

| n/a | Confirmed | |
|---|---|---|
| ☐ | ☒ | The exact sample size (*n*) for each experimental group/condition, given as a discrete number and unit of measurement |
| ☐ | ☒ | A statement on whether measurements were taken from distinct samples or whether the same sample was measured repeatedly |
| ☐ | ☒ | The statistical test(s) used AND whether they are one- or two-sided<br>*Only common tests should be described solely by name; describe more complex techniques in the Methods section.* |
| ☐ | ☒ | A description of all covariates tested |
| ☐ | ☒ | A description of any assumptions or corrections, such as tests of normality and adjustment for multiple comparisons |
| ☐ | ☒ | A full description of the statistical parameters including central tendency (e.g. means) or other basic estimates (e.g. regression coefficient) AND variation (e.g. standard deviation) or associated estimates of uncertainty (e.g. confidence intervals) |
| ☐ | ☒ | For null hypothesis testing, the test statistic (e.g. *F*, *t*, *r*) with confidence intervals, effect sizes, degrees of freedom and *P* value noted<br>*Give P values as exact values whenever suitable.* |
| ☒ | ☐ | For Bayesian analysis, information on the choice of priors and Markov chain Monte Carlo settings |
| ☒ | ☐ | For hierarchical and complex designs, identification of the appropriate level for tests and full reporting of outcomes |
| ☒ | ☐ | Estimates of effect sizes (e.g. Cohen's *d*, Pearson's *r*), indicating how they were calculated |

*Our web collection on statistics for biologists contains articles on many of the points above.*

## Software and code

Policy information about availability of computer code

| Data collection | There was no software used in data collection. |
|---|---|
| Data analysis | We used the publicly available software plink (v2.0; https://www.cog-genomics.org/plink/2.0/) and R (v4.4.3). We used the R packages sffdr (v1.0.0; https://github.com/ajbass/sffdr), CAMT (v1.1; https://github.com/jchen1981/CAMT), adaptMT (v1.0.0), swfdr (v1.34.0), qvalue (v2.38.0), tidyverse (v2.0.0), patchwork (v1.3.0), and coloc (v5.2.3). The code to reproduce the results in the manuscript is available at https://github.com/ajbass/sffdr_manuscript. |

For manuscripts utilizing custom algorithms or software that are central to the research but not yet described in published literature, software must be made available to editors and reviewers. We strongly encourage code deposition in a community repository (e.g. GitHub). See the Nature Portfolio guidelines for submitting code & software for further information.

## Data

Policy information about availability of data

All manuscripts must include a data availability statement. This statement should provide the following information, where applicable:

- Accession codes, unique identifiers, or web links for publicly available datasets
- A description of any restrictions on data availability
- For clinical datasets or third party data, please ensure that the statement adheres to our policy

The asthma (GCST007800, GCST007799), eosinophil count (GCST004606), rheumatoid arthritis (GCST002318), hypothyroidism (GCST90013893) and EGPA

## Human research participants

Policy information about studies involving human research participants and Sex and Gender in Research.

| | |
|---|---|
| Reporting on sex and gender | All analyses included males and females. We report that sex was included as a covariate in association analyses. |
| Population characteristics | The average age of the UK Biobank is 57 where 54% of participants are female. The average age of the MVP Biobank is 67 where 8% are female. The median age of FinnGen Biobank is 53 where 57% are female. |
| Recruitment | The UK Biobank recruited half a million volunteers 40–69 years old in the United Kingdom across 22 assessment centers from 2006 to 2010. See https://www.ukbiobank.ac.uk/media/gnkeyh2q/study-rationale.pdf for additional details. The FinnGen Biobank recruited over 500,000 individuals (release 12) and the MVP Biobank enrolled over 450,000 (version 11) individuals. See https://www.finngen.fi/en/node/1985 and https://www.ncbi.nlm.nih.gov/projects/gap/cgi-bin/study.cgi?study_id=phs001672.v11.p1 for recruitment details for FinnGen and MVP Biobanks, respectively. |
| Ethics oversight | The research in this work abides by the Helsinki Declaration. The North West Multi-centre Research Ethics Committee (MREC) approved the protocols of the UK Biobank study. The REC reference number is 06/MRE08/65. The FinnGen Biobank collected informed consent for research via the Finnish Biobank Act. The Central VA Institutional Review Board (IRB) provided appropriate consent and protocol approvals for the MVP Biobank. |

Note that full information on the approval of the study protocol must also be provided in the manuscript.

# Field-specific reporting

Please select the one below that is the best fit for your research. If you are not sure, read the appropriate sections before making your selection.

☒ Life sciences  ☐ Behavioural & social sciences  ☐ Ecological, evolutionary & environmental sciences

For a reference copy of the document with all sections, see nature.com/documents/nr-reporting-summary-flat.pdf

# Life sciences study design

All studies must disclose on these points even when the disclosure is negative.

| | |
|---|---|
| Sample size | No sample size calculation was performed. We analyzed available participants in the UK Biobank (sample size: 380,600). The summary statistics for the MVP, FinnGen, asthma, eosinophil count, EGPA, rheumatoid arthritis, and hypothyroidism studies were previously published. |
| Data exclusions | In the UK Biobank study, there were a few individuals that withdrew consent. We removed individuals of non-British descent and any individuals that were related. This information was provided by the UK Biobank. We also removed individuals with a sex chromosome aneuploidy, and genotype missing rate >0.05. After filtering, there were 380,600 individuals in our analysis. |
| Replication | We searched previous GWAS findings of immune-related traits to determine whether significant loci were previously known. There was no experimental replication attempted. |
| Randomization | The data used was from observation studies and so randomization was not applicable to our study. We regressed out covariates such as sex, age, and the top 20 PCs to account for ancestry in the UK Biobank study. The other studies were also observational studies where ancestry and other observed covariates were accounted for in the testing procedure. |
| Blinding | Blinding was not applicable in our study as we used coded de-identified data. |

# Reporting for specific materials, systems and methods

We require information from authors about some types of materials, experimental systems and methods used in many studies. Here, indicate whether each material, system or method listed is relevant to your study. If you are not sure if a list item applies to your research, read the appropriate section before selecting a response.

## Materials & experimental systems

| n/a | Involved in the study |
|---|---|
| ☒ | ☐ Antibodies |
| ☒ | ☐ Eukaryotic cell lines |
| ☒ | ☐ Palaeontology and archaeology |
| ☒ | ☐ Animals and other organisms |
| ☒ | ☐ Clinical data |
| ☒ | ☐ Dual use research of concern |

## Methods

| n/a | Involved in the study |
|---|---|
| ☒ | ☐ ChIP-seq |
| ☒ | ☐ Flow cytometry |
| ☒ | ☐ MRI-based neuroimaging |

