## [Peer Review File · Nature Computational Science]

Exploiting pleiotropy to enhance variant discovery with functional false discovery rates

Corresponding Author: Dr Andrew Bass

Version 0:

Decision Letter:

**** Please ensure you delete the link to your author homepage in this e-mail if you wish to forward it to your co-authors. ****

Dear Dr Bass,

Your manuscript "Exploiting pleiotropy to enhance variant discovery with functional false discovery rates" has now been seen by 3 referees, whose comments are appended below. You will see that while they find your work of interest, they have raised points that need to be addressed before we can make a decision on publication.

The referees' reports seem to be quite clear. Naturally, we will need you to address **all** of the points raised.

While we ask you to address all of the points raised, the following points need to be substantially worked on:

- Please discuss prior methods that leverage pleiotropy to enhance statistical power.
- The manuscript introduces several empirical strategies in the Methods section, and their performance should be thoroughly demonstrated or discussed.
- The complex structure of linkage disequilibrium (LD) is a key challenge in applying FDR to GWAS. Please extend the sfFDR framework to account for SNPs in LD is crucial.
- Unlike MTAG, the sfFDR framework does not output SNP-level effect sizes, which may limit its application in certain post-GWAS analyses.
- Using simulations, please explicitly address rare disease scenarios.
- The simulation should consider more realistic LD scenarios.
- In the simulation settings, please also consider the scenarios when the causal variants are not included by the genotyped SNPs.
- In the benchmark analyses, include comparisons with established summary-level multi-trait methods, particularly MTAG (Turley et al., Nature Genetics 2018) and similar approaches.
- For the novel EGPA signals identified by sfFDR, validation is crucial and should be added.
- Discuss how different genetic architectures (e.g., polygenicity levels, effect size distributions across allele frequencies) might impact sfFDR performance.
- Please report runtime and memory requirements for typical GWAS summary statistics with different number of variants and/or conditioning GWASs, ideally in comparison to other existing methods.

In addition to these points, while the computational cost for the case study is mentioned, a direct comparison against the standard workflow should be provided.

Please use the following link to submit your revised manuscript and a point-by-point response to the referees' comments (which should be in a separate document to any cover letter):

Link Redacted

**** This url links to your confidential homepage and associated information about manuscripts you may have submitted or be reviewing for us. If you wish to forward this e-mail to co-authors, please delete this link to your homepage first. ****

To aid in the review process, we would appreciate it if you could also provide a copy of your manuscript files that indicates your revisions by making use of Track Changes or similar mark-up tools. Please also ensure that all correspondence is marked with your Nature Computational Science reference number in the subject line.

In addition, please make sure to upload a Word Document or LaTeX version of your text, to assist us in the editorial stage.

To improve transparency in authorship, we request that all authors identified as 'corresponding author' on published papers create and link their Open Researcher and Contributor Identifier (ORCID) with their account on the Manuscript Tracking System (MTS), prior to acceptance. ORCID helps the scientific community achieve unambiguous attribution of all scholarly contributions. You can create and link your ORCID from the home page of the MTS by clicking on 'Modify my Springer Nature account'. For more information please visit <http://www.springernature.com/orcid>.

We hope to receive your revised paper within three weeks. If you cannot send it within this time, please let us know.

Best regards,

Ananya Rastogi, PhD
Senior Editor
Nature Computational Science

Reviewers comments:

Reviewer #1 (Remarks to the Author):

This paper proposes a method to leverage pleiotropy to enhance statistical power. Based on Bayesian theory, the method outputs both FDR-related q-values and regular p-values. The general idea is straightforward, theoretically sound, and potentially useful. While most GWAS studies rely on Bonferroni correction to control type I error rates, I am pleased to see more discussions around FDR in this context. However, I believe the paper requires revisions to better demonstrate its effectiveness and advantages over existing methods. Detailed comments are provided below:

1. The paper does not sufficiently discuss prior methods that leverage pleiotropy to enhance statistical power. For instance, MTAG utilizes summary statistics from multiple traits to improve GWAS for a single trait. Although this paper additionally outputs FDR-related q-values, the ultimate goal of MTAG is quite similar to that of this method. These similar methods should also be mentioned in the Discussion section and compared in the simulation studies and real data analyses. Without demonstrating a substantial advantage over existing methods, such as MTAG, the general interest in this method may be limited.
2. As a methodological paper, it introduces several empirical strategies in the Methods section, but their performance is not thoroughly demonstrated or discussed. For example, the paper defines $\pi_0(z)$ as the probability that SNPs are null given z . If the original p-values closely follow a uniform distribution, $\pi_0(z)$ should be very close to zero for most values of z . However, it is unclear: 1) Whether this assumption is correct, 2) Whether the estimation of $\pi_0(z)$ is robust, and 3) Whether this estimation could significantly affect the final results. These issues should be addressed through extensive simulation studies and real data analyses, which would greatly enhance understanding of the method for both data analysis and future methodological development.
3. A more detailed discussion of $f(p|z)$ and its related surrogate variable approximation is necessary, as highlighted in comment #2. This would provide greater clarity on the assumptions and implications of the method.
4. The complex structure of linkage disequilibrium (LD) is a key challenge in applying FDR to GWAS. Extending the sfFDR framework to account for SNPs in LD is crucial. In Section 4.7, the paper provides a real data example to demonstrate how SNPs in LD can be excluded. However, more discussion on the selection strategy is needed. For instance, if SNPs with lower p-values for informative traits are preferentially selected, could this inflate the results? Addressing this concern would strengthen the paper.
5. Recent studies have shown that MTAG may result in inflation when its core assumptions are violated. I am concerned that the sfFDR framework might face similar issues. I suggest adding a real data analysis similar to the EGPA study mentioned in the paper. In this analysis, the p-values for informative traits should remain the same, while the standard p-values are simulated following a uniform distribution. The sfFDR framework can then be applied to calculate functional p-values. If the resulting p-values still follow a uniform distribution, this would provide strong evidence that the method can well control type I error rates. Conversely, if SNPs associated with the informative traits are identified, it would indicate that the sfFDR framework fails to control type I error rates.
6. Following up on comment #5, the corresponding $\pi_0(z)$ and $f(p|z)$ values from both the original analysis and the suggested analysis should be discussed. This would help explain the results and provide further insights into the robustness of the method.
7. Unlike MTAG, the sfFDR framework does not output SNP-level effect sizes, which may limit its application in certain post-GWAS analyses. However, I do not think this limitation largely detracts from the novelty of the paper, provided the above comments are adequately addressed. Additionally, I appreciate the paper's focus on FDR-related work, though I remain concerned about how the LD structure might affect performance.

Reviewer #2 (Remarks to the Author):

Summary:

In this manuscript, Bass et al. present sfFDR that leverages information from related traits to boost statistical power in genome-wide association studies (GWAS). The authors demonstrate the method's effectiveness through simulations and applications to both common traits and a rare disease study, showing substantial gains in power equivalent to that from increasing sample size. The method and results appear to represent a useful contribution to the field. However, several major issues need to be addressed before the manuscript is suitable for publication.

1. In the main text (line 108-109), please specify what values "High", "Medium" and "Low" represents and why these values are chosen. Currently, these values are only mentioned on line 532 and no justifications are made on why they are chosen as such. Similarly, on line 110, please specify why 1.25% and 2.50% are chosen as the number of overlapped non-null SNPs.
2. While the authors analyze a rare disease (EGPA) in their real data application, the simulations don't explicitly address rare disease scenarios. The authors should either clarify if the 'Low' scenario is meant to represent rare diseases, or add simulations specifically modeling rare disease characteristics (e.g., low prevalence, potentially larger effect sizes).
3. The simulation should consider more realistic LD scenarios. Currently, the analysis uses UK Biobank subsets, but in practice, primary and informative GWASs often come from different cohorts with slightly different LD patterns. The authors should simulate these cross-cohort LD differences to evaluate method robustness.
4. In the simulation settings, the authors should also consider the scenarios when the causal variants are not included by the genotyped SNPs. Often in real-world scenarios, the causal variant is untyped and the genotyped SNPs are partially tagging it. The authors should investigate (or at least comment on) how their method would behave under such circumstances.
5. In Supplementary Figure 8, while the median empirical FDRs at level 0.01 are approximately 0.01, all methods show long tails in their distributions. Does this suggest poorer FDR calibration in the presence of LD? The authors should discuss the implications for practical applications.
6. In the benchmark analyses, the authors should include comparisons with established summary-level multi-trait methods, particularly MTAG (Turley et al., Nature Genetics 2018) and similar approaches. While these methods use different statistical frameworks, they share the core goal of leveraging summary statistics from related traits to enhance GWAS power. A systematic comparison would help users understand when to prefer sfFDR over existing methods.
7. In the BMI analysis, the authors only tested scenarios where conditioning traits are either all informative or all uninformative. However, in real practice, users might have a mix of both types when selecting traits. The authors should evaluate how sfFDR performs with a mixture of informative and uninformative conditioning traits, particularly how the proportion of uninformative traits affects power. This would provide practical guidance for users who may not be able to perfectly select relevant traits.
8. For the novel EGPA signals identified by sfFDR, validation is crucial. The authors should either attempt replication in independent EGPA cohorts, or demonstrate through simulations (relevant to point 2 above). This is particularly important given sfFDR's potential utility for rare disease studies.
9. More concrete guidelines for selecting informative traits would be valuable, particularly for rare diseases. Can metrics like genetic correlation estimates guide trait selection? The authors should also clarify how they define 'high-powered' traits (line 293) - is this based on significant variant count, heritability estimates, or other metrics?
10. The authors should discuss how different genetic architectures (e.g., polygenicity levels, effect size distributions across allele frequencies) might impact sfFDR performance, even if the method is theoretically robust to these factors.
11. The authors should report runtime and memory requirements for typical GWAS summary statistics with different number of variants and/or conditioning GWASs, ideally in comparison to other existing methods.

Reviewer #2 (Remarks on code availability):

The software package is available on CRAN (<https://github.com/ajbass/sffdr>) with clear installation instructions and a working vignette. The installation process and running the example code were smooth.

Reviewer #3 (Remarks to the Author):

This manuscript proposes a novel method to exploit currently published GWAS summary statistics related to the main trait

under investigation. By using the surrogate functional false discovery rate (sfFDR), the authors show that their method increases power to discover new loci associated to the main trait by leveraging the summary statistics coming from other related traits. Furthermore, the method outputs a functional p-value and Bayes factor that could help fine-mapping and colocalization. Simulations show that their method appropriately controls for FDR and Type 1 error while increasing power over standard p-values. Also, one of the strengths of their method is that it allows either the use of the FDR (more popular in eQTL studies) or FWER (more popular in GWAS) control since there is a direct mapping between the functional q-values and functional p-values.

The manuscript is well-written, easy to follow and the Methods section not too difficult to read even for a non-statistics audience. The Abstract, Introduction and Discussion sections are clear. The reference list is complete and appropriate. I only have a few comments:

1. I found a somewhat "strange" behavior in Figure 2b and Supp Fig 6. When the effect size of informative studies is "Moderate", the number of discoveries is always lower than when the effect size is "None". Could the authors explain why?
2. Is there any prior studies or literature suggesting that the overlap of shared non-null SNPs (i.e. pleiotropy) between informative studies and the primary study is around 1.25% or 2.50%? Can it be modified by the user when using the software?
3. Somehow related to point 2 above, is there any recommendation as to the number of informative studies the user should include before running the method? Does this choice be guided by moderate-to-strong genetic correlations between the traits?

Small typos:

Line 201: "... of EGPAS is unknown ..."

Line 226: "... there were 226 discoveries ..."

In summary, this novel method will be of interest to many practitioners in the GWAS community.

Reviewer #3 (Remarks on code availability):

I tried to install the package but the "qvalue" package was not available in the latest version of R (4.4.2):

```
"Warning in install.packages :  
package 'qvalue' is not available for this version of R"
```

The qvalue package needs to be updated first before running sfldr.

Version 1:

Decision Letter:

Our ref: NATCOMPUTSCI-24-2181A

17th April 2025

Dear Dr. Bass,

Thank you for submitting your revised manuscript "Exploiting pleiotropy to enhance variant discovery with functional false discovery rates" (NATCOMPUTSCI-24-2181A). It has now been seen by the original referees and their comments are below. The reviewers find that the paper has improved in revision, and therefore we'll be happy in principle to publish it in Nature Computational Science, pending minor revisions to satisfy the referees' final requests and to comply with our editorial and formatting guidelines.

TRANSPARENT PEER REVIEW

Nature Computational Science offers a transparent peer review option for original research manuscripts. We encourage increased transparency in peer review by publishing the reviewer comments, author rebuttal letters and editorial decision letters if the authors agree. Such peer review material is made available as a supplementary peer review file. **Please remember to choose, using the manuscript system, whether or not you want to participate in transparent peer review.**

Please note: we allow redactions to authors' rebuttal and reviewer comments in the interest of confidentiality. If you are concerned about the release of confidential data, please let us know specifically what information you would like to have removed. Please note that we cannot incorporate redactions for any other reasons. Reviewer names will be published in the peer review files if the reviewer signed the comments to authors, or if reviewers explicitly agree to release their name. For

more information, please refer to our [FAQ page](https://www.nature.com/documents/nr-transparent-peer-review.pdf).

Thank you again for your interest in Nature Computational Science. Please do not hesitate to contact me if you have any questions.

Sincerely,

Ananya Rastogi, PhD
Senior Editor
Nature Computational Science

ORCID

Reviewer #1 (Remarks to the Author):

Comments have been addressed.

Reviewer #2 (Remarks to the Author):

I would like to thank the authors for fully addressing my comments. I have no further comments and hope to see the manuscript out soon.

Reviewer #3 (Remarks to the Author):

The authors satisfactorily addressed my comments.

Furthermore, based on comments raised by the other 2 Reviewers, they added simulation results comparing their method to MTAG, and simulations demonstrating their method's robustness against non-informative traits and different LD structure between the primary trait and correlated (informative) traits. The case study presented using real data (EGPA + 4 additional rare diseases) was convincing enough that the method could detect novel variants.

As a result, the manuscript has greatly improved.

Reviewer #3 (Remarks on code availability):

I was able to install the code on my machine using RStudio. I did not run it though, but the GitHub page shows how to reproduce an example based on small GWAS summary statistics datasets.

Version 2:

Decision Letter:

Dear Dr Bass,

We are pleased to inform you that your Article "Exploiting pleiotropy to enhance variant discovery with functional false discovery rates" has now been accepted for publication in Nature Computational Science.

Once your manuscript is typeset, you will receive an email with a link to choose the appropriate publishing options for your paper and our Author Services team will be in touch regarding any additional information that may be required.

Authors may need to take specific actions to achieve compliance with funder and institutional open access mandates.

If your research is supported by a funder that requires immediate open access (e.g. according to [Plan S principles](https://www.springernature.com/gp/open-science/plan-s-compliance) or the [NIH public access policy](https://www.springernature.com/gp/open-science/us-federal-agency-compliance)) then you should select the gold OA route, and we will direct you to the compliant route where possible. Because authors warrant under our subscription licensing terms that they haven't committed to licensing any version of their article under a licence inconsistent with the terms of our agreement – including the applicable embargo period – publication under the subscription model isn't suitable for authors whose funders require no embargo.

Acceptance of your manuscript is conditional on all authors' agreement with our publication policies (see <https://www.nature.com/natcomputsci/for-authors>). In particular your manuscript must not be published elsewhere and there must be no announcement of the work to any media outlet until the publication date (the day on which it is uploaded onto our web site).

Before your manuscript is typeset, we will edit the text to ensure it is intelligible to our wide readership and conforms to house style. We look particularly carefully at the titles of all papers to ensure that they are relatively brief and understandable.

Once your manuscript is typeset, you will receive a link to your electronic proof via email with a request to make any corrections within 48 hours. If, when you receive your proof, you cannot meet this deadline, please inform us at rjsproduction@springernature.com immediately.

If you have queries at any point during the production process then please contact the production team at rjsproduction@springernature.com.

We welcome the submission of potential cover material (including a short caption of around 40 words) related to your manuscript; suggestions should be sent to Nature Computational Science as electronic files (the image should be 300 dpi at 210 x 297 mm in either TIFF or JPEG format). We also welcome suggestions for the Hero Image, which appears at the top of our [home page](http://www.nature.com/natcomputsci); these should be 72 dpi at 1400 x 400 pixels in JPEG format. Please note that such pictures should be selected more for their aesthetic appeal than for their scientific content, and that colour images work better than black and white or grayscale images. Please do not try to design a cover with the Nature Computational Science logo etc., and please do not submit composites of images related to your work. I am sure you will understand that we cannot make any promise as to whether any of your suggestions might be selected for the cover of the journal.

Best regards,

Jie Pan, Ph.D. (on behalf of Ananya)
Senior Editor
Nature Computational Science

P.S. Click on the following link if you would like to recommend Nature Computational Science to your librarian: <https://www.springernature.com/gp/librarians/recommend-to-your-library>

** Visit the Springer Nature Editorial and Publishing website at <http://editorial-jobs.springernature.com> for more information about our career opportunities. If you have any questions please click [here](mailto:editorial.publishing.jobs@springernature.com). **

Response to Referees

We thank the reviewers for their constructive comments on our manuscript, which have substantially enhanced the quality of our revised work. We have responded to concerns raised by the reviewers below. The review has been included in its entirety (with minor formatting changes), and we have inserted our responses in the shaded boxes after the relevant paragraph. We have formatted the revised manuscript so that **new text is in red**, although moved text remains in black.

Reviewer 1

This paper proposes a method to leverage pleiotropy to enhance statistical power. Based on Bayesian theory, the method outputs both FDR-related q-values and regular p-values. The general idea is straightforward, theoretically sound, and potentially useful. While most GWAS studies rely on Bonferroni correction to control type I error rates, I am pleased to see more discussions around FDR in this context. However, I believe the paper requires revisions to better demonstrate its effectiveness and advantages over existing methods. Detailed comments are provided below:

Comments

1. The paper does not sufficiently discuss prior methods that leverage pleiotropy to enhance statistical power. For instance, MTAG utilizes summary statistics from multiple traits to improve GWAS for a single trait. Although this paper additionally outputs FDR-related q-values, the ultimate goal of MTAG is quite similar to that of this method. These similar methods should also be mentioned in the Discussion section and compared in the simulation studies and real data analyses. Without demonstrating a substantial advantage over existing methods, such as MTAG, the general interest in this method may be limited.

We thank the reviewer for their suggestion and have added extensive simulations comparing sfFDR to MTAG (Figure 3,S18). In our UK Biobank study, we found that MTAG did not lead to any power improvements compared to the standard analysis when leveraging body fat percentage (BFP), triglycerides, and cholesterol. As MTAG can have poor performance when applied to many traits [1, 2], we considered a simpler scenario where we only used the most informative trait, BFP, in the analysis. In this case, MTAG substantially increases the number of discoveries but with a much lower replication rate, implying that a substantial number of discoveries are likely false discoveries.

We also considered MTAG in the context of our new null simulations using the UK Biobank study (see our response to comment #5). We found that MTAG can inflate the type I error rate. We believe that these new results will help distinguish our approach from MTAG while also increasing the general interest and usefulness of sfFDR.

We have added the MTAG comparisons in the Results section:

We then compared sfFDR to an analysis that uses MTAG [1] and a standard GWAS analysis of the primary trait BMI. The number of discoveries from sfFDR was substantially larger than the standard GWAS analysis across a range of sample sizes (Figure 3a). Furthermore, nearly all of the discoveries made with the p_f -values from sfFDR were found by a meta-analysis of both data sets (BMI only; Figure 3b), suggesting that the additional discoveries are a subset of those that would be found by increasing the sample size. When applying MTAG to the obesity-related traits, we found that MTAG did not increase the number of discoveries and performed worse than the standard analysis at some sample sizes. As MTAG can have poor performance when applied to many traits [1, 2], we considered a simpler scenario where we only used the most informative trait, BFP, in the analysis. In this case, MTAG substantially increased the number of discoveries compared to sfFDR and the standard analysis but with a much lower replication rate (particularly at lower sample sizes; Figure 3b), implying that a substantial number of discoveries are likely false discoveries. In contrast, the vast majority of discoveries by sfFDR were

replicated in the meta-analysis approach. Thus, these results demonstrate the potential of sfFDR to substantially increase the power in GWAS studies by leveraging related traits.

Please see our response to comment #5 for detailed comparisons of MTAG and sfFDR under the null setting.

2. As a methodological paper, it introduces several empirical strategies in the Methods section, but their performance is not thoroughly demonstrated or discussed. For example, the paper defines $\pi_0(z)$ as the probability that SNPs are null given z . If the original p-values closely follow a uniform distribution, $\pi_0(z)$ should be very close to zero for most values of z . However, it is unclear: 1) Whether this assumption is correct, 2) Whether the estimation of $\pi_0(z)$ is robust, and 3) Whether this estimation could significantly affect the final results. These issues should be addressed through extensive simulation studies and real data analyses, which would greatly enhance understanding of the method for both data analysis and future methodological development.

To demonstrate the performance of the $\pi_0(z)$ estimator, we have added new results to show that sfFDR's $\pi_0(z)$ estimator achieves the smallest root mean square error (RMSE) across all of the FDR methods in our simulation study (Figure S4). We note that the chosen $\pi_0 = 0.98$ in the simulations represents a sparse signal setting expected for rarer diseases. In particular, this value reflects the π_0 value estimated in the EGPA study ($\pi_0 = 0.982$; see Figure S1 for the primary and informative p-value comparisons between the primary analysis and simulations). In this case, we found that sfFDR consistently provides conservative estimates of π_0 which is theoretically expected (see Methods section). As such, this leads to conservative estimates of the underlying local FDRs, q-values, and functional p-values.

We implemented a new set of simulations when $\pi_0 = 1$ (i.e., all SNPs are null and so the p-values are uniformly distributed) and found that the sfFDR's $\pi_0(z)$ estimate still had the lowest RMSE among competitors (Figure S26). However, we observed a slight

bias when $\pi_0 = 1$ due to overfitting (Figures S27), and a consequence of this is a slight upward bias in the type 1 error rate when calculating the functional p-value in the $\pi_0 = 1$ setting (Figure S19). This is not a theoretical bias but an estimation one (i.e., overfitting): when we assign the proportion of truly null tests to be the average (i.e., constant) and set the nearest neighbor parameter in the kernel density estimate to be 0.7, the bias is minimal. As such, due to the reviewers comments, we have decided to add a test in the software to catch the global null case, warn the user, and default the estimator as outlined above. Users can also use LD score regression to identify the global null case of their p-values similar to MTAG. We do not recommend users use the package in the global null setting as there will be no power improvements.

We note that we evaluated the $\pi_0 = 1$ case further using the UK Biobank data and found that the inflation is slight and did not lead to any false discoveries at the genome-wide significance threshold while MTAG can lead to substantially inflated p-values (Figure S18). Importantly, we do not observe any bias in simulations or when using real data when there is a signal in the primary study. Alternatively, MTAG can still lead to substantially inflated p-values of the null SNPs (see further details in response to question #5 below).

We added these findings in the Results section:

We compared the sfFDR framework to other FDR procedures that can incorporate multiple informative variables, namely, AdaPT [3], CAMT [4], and an estimator by Boca et al. (2018; referred to as the “Boca-Leek” method) [5]. When evaluating the root mean square error (RMSE) of the estimated $\pi_0(z)$, we found that sfFDR achieves the smallest RMSE across all methods (Figure S4). We then compared estimates of the proportion of truly null hypotheses (i.e., the average value of $\pi_0(z)$) and found that CAMT was anti-conservative (predicts more non-null SNPs than exist), AdaPT and Boca-Leek were slightly anti-conservative, and sfFDR was conservative (Figure S5). Note that a conservative estimator is more desirable than an anti-conservative estimator as it will not overestimate the amount of signal (which can lead to an inflated FDR).

We added a discussion of the $\pi_0 = 1$ case when evaluating MTAG and sfFDR in the UK Biobank analysis (see response to comment #5).

3. A more detailed discussion of $f(p|z)$ and its related surrogate variable approximation is necessary, as highlighted in comment #2. This would provide greater clarity on the assumptions and implications of the method.

We added new results to evaluate the RMSE of the log-transformed density $f(p|z)$ (Figure S6). We found that sfFDR's surrogate density estimator outperformed the marginal density estimate (Q-value) and AdaPT, but did not outperform CAMT (a parametric model). Coincidentally, our data was generated in such a way that matches the assumptions behind CAMT's parametric model, and so we can view CAMT as a baseline parametric model comparisons. As such, it is unsurprising that it outperformed a non-parametric estimator. More importantly, when we compared the functional q-values, which in turn depend on the local FDR (i.e., $\pi_0(z)$ and $f(p|z)$), we found that it slightly outperformed CAMT (and substantially outperformed the other procedures) while getting close to the oracle procedure (i.e., calculated with the true values). Thus, we find that the surrogate approximation from sfFDR has improved performance compared to other FDR methods in our setting.

We added these changes to the Results section:

Since sfFDR relies on a non-parametric surrogate density estimator to handle many informative variables, we compared the performance of the surrogate density estimator to AdaPT and CAMT. While sfFDR's estimated surrogate density had a smaller RMSE of the log-transformed density values compared to AdaPT and to a marginal density estimator, it did not outperform CAMT (Figure S6). However, our data generating process matched the assumptions behind CAMT's parametric model, and so it is unsurprising CAMT's parametric density estimator outperformed sfFDR's non-parametric density estimator. In general, a non-parametric density estimator offers more flexibility by not assuming an underlying distribution of the data.

When we compared the corresponding estimated q_f -values, which in turn depend on the local FDR (i.e., $\pi_0(z)$ and $f(p|z)$), we found that these methods controlled the FDR (Figure S2), although the FDR was inflated for CAMT when the primary study power was “Low” (Figure S7). Furthermore, sfFDR and CAMT had comparable power and outperformed AdaPT and Boca-Leek across a range of small FDR thresholds (Figure S7). Overall, we found that sfFDR—and thus the surrogate density approximation—had improved performance compared to other FDR procedures in our setting.

4. The complex structure of linkage disequilibrium (LD) is a key challenge in applying FDR to GWAS. Extending the sfFDR framework to account for SNPs in LD is crucial. In Section 4.7, the paper provides a real data example to demonstrate how SNPs in LD can be excluded. However, more discussion on the selection strategy is needed. For instance, if SNPs with lower p-values for informative traits are preferentially selected, could this inflate the results? Addressing this concern would strengthen the paper.

We expanded our simulations to include different SNP selection strategies to train sfFDR. There were two primary types of selection strategies considered, namely, haplotype blocks and pruning. For the haplotype block setting, we partitioned the genome into independent (or uncorrelated) blocks and either randomly sampled a representative SNP or selected a representative SNP with the smallest informative trait p-value. In the pruned SNP setting, we applied PLINK with squared correlation thresholds of 0.05, 0.1, 0.2, 0.3, and 0.4 (calculated on a sliding window of the 1,000 closest SNPs). This resulted in training sfFDR on 113,834, 180,915, 272,351, 336,932, and 380,552 SNPs. Finally, we considered training sfFDR on all the SNPs in the UK Biobank study.

Surprisingly, we found marginal differences in terms performance in the UK Biobank (Figure S13). In fact, training sfFDR on all the SNPs had the best performance. Notably, in our new null simulations (see response in comment #5), the functional p-values were similar across the different SNP selection strategies (Figure S20). Because our primary application is rarer diseases, we selected SNPs based on the haplotype block

setting with informative trait sampling to maximize coverage of pleiotropic SNPs. For more common diseases, our results suggest there is much more flexibility on the set of SNPs used to train sfFDR and, in such cases, the primary limitation is computational time (Figure S14 shows computational time).

We added these changes to the Results section:

We first evaluated the performance of two SNP selection strategies to train sfFDR, namely, selecting SNPs within haplotype blocks or through pruning. For the haplotype block setting, we partitioned the genome into independent (or uncorrelated) blocks and selected a representative SNP either randomly or with the smallest informative trait p -value. When comparing the number and replicability of discoveries with a meta-analysis of both data sets (BMI only), we found marginal differences between pruning at multiple correlation thresholds and haplotype block sampling (Figure S13). Since our primary application is rare diseases, we selected SNPs based on haplotype blocks with informative trait sampling to maximize coverage of pleiotropic SNPs and minimize computation time (Figure S14).

and the Discussion section:

First, since our primary application is rare diseases, we trained sfFDR using haplotype blocks with informative trait sampling to maximize coverage of pleiotropic SNPs. However, for more common diseases, our results suggest there is flexibility in how to choose SNPs (e.g., using high pruning correlation thresholds or potentially using all of the SNPs) and, in such cases, the primary limitation is computational time (Figure S14).

and the Methods section:

There were two types of SNP selection strategies considered in this work, namely, using haplotype blocks and pruning. For the haplotype block setting, LD-independent SNPs were determined by using pre-defined haplotype blocks constructed using the LDAK method [6]. More specifically, within each

haplotype block, we performed hierarchical clustering using a random subset of 40,000 individuals from the UK Biobank to identify clusters of uncorrelated SNPs. In total, there were 161,207 “independent” clusters at a pruned correlation threshold of 0.99. At each cluster, a representative SNP was selected either randomly or based on having the smallest informative trait p -value (see Section 4.8). In the pruned SNP setting, we applied PLINK varying the squared correlation threshold to 0.05, 0.1, 0.2, 0.3, and 0.4 (calculated on a sliding window of the 1,000 closest SNPs). This resulted in training sfFDR on 113,834, 180,915, 272,351, 336,932, and 380,552 SNPs. Additionally, we also used all of the SNPs to train sfFDR. Finally, in our implementation of the GAM model, we fit a natural cubic spline to the informative traits p -values with knots placed at the 0.005, 0.025, 0.01, 0.05, 0.1 quantiles.

5. Recent studies have shown that MTAG may result in inflation when its core assumptions are violated. I am concerned that the sfFDR framework might face similar issues. I suggest adding a real data analysis similar to the EGPA study mentioned in the paper. In this analysis, the p -values for informative traits should remain the same, while the standard p -values are simulated following a uniform distribution. The sfFDR framework can then be applied to calculate functional p -values. If the resulting p -values still follow a uniform distribution, this would provide strong evidence that the method can well control type I error rates. Conversely, if SNPs associated with the informative traits are identified, it would indicate that the sfFDR framework fails to control type I error rates.

It is important to first note that MTAG can be unreliable in certain settings, as noted by the original authors [1] and others (see, e.g., [2]). The authors of MTAG mention that bias results when a SNP is null in one trait and non-null in the other traits. More precisely, the model assumes that the correlation between effects across traits is the same for all SNPs, leading to cases where the p -values can be inflated from MTAG. While this may be a reasonable assumption when applied to different summary statistics of the same trait, we show that it can lead to inflated p -values when the genetic architecture varies

across traits.

Alternatively, sfFDR does not make this strict assumption and has theoretical control of the type I error rate in such settings. Specifically, we define the functional p-value as

$$p_f(p, z) = \inf_{\{\Gamma_\tau: t \in \Gamma_\tau\}} \Pr(T \in \Gamma_\tau \mid H = 0), \quad (1)$$

where the variable definitions can be found in the Methods section. In the case where the p-values are uniformly distributed, the q-values (and local FDR) of the SNPs are 1 and the above equation simplifies to

$$p_f(p, z) = \Pr(T \in \Gamma_{\Lambda(p)}), \quad (2)$$

which is the distribution function of the local FDRs of the original p-values. Theoretically, this value is $p_f(p, z) = 1$ and, in such cases, we can accomplish type I error rate control at level α via randomization (i.e., breaking ties randomly), and so the functional p-value will be uniformly distributed. While any ranking is theoretically valid, we base it on the ranking of the original p-values.

We have added extensive simulations to address the reviewers concerns by considering two new null settings (Figure 3, S20):

- UK Biobank scenario #1: We randomly sampled 3 chromosomes from the UK Biobank BMI summary statistics. We then assigned the unselected BMI chromosomes to be the null summary statistics from permuting the traits. We designed the simulation this way so that we knew which SNPs were non-null while maintaining the genetic correlation at the three selected chromosomes. In this case, we found that sfFDR's p-values are well-behaved while MTAG p-values are inflated when conditioning on BFP (Figure 3d). We note this behavior is expected because MTAG assumes that the covariance of true SNP effects is the same across all SNPs, otherwise the null SNPs get pulled into significance (this behavior has been pointed out by the original MTAG authors [1]).
- UK Biobank scenario #2: We used permuted null BMI summary statistics and the original (non-null) summary statistics of the informative traits. In this case, the proportion of truly null tests in BMI is 1 and there is no genetic signal. Once again, we

observed a substantial inflation of MTAG's p -values (Figure S18). We do note that MTAG attempts to identify this case (i.e., if any trait looks null) and stops the software from running and so we had to use the 'force' setting. Alternatively, sfFDR's functional p -value exhibits nearly uniform p -values but we do identify a slight bias. The bias is not due to any theoretical issues with sfFDR but because of overfitting (see response to comment #2). Even though we observe that the bias did not lead to any newly discovered SNPs (unlike MTAG), we added a note to users using the software to identify the uniform p -value case (either using an LD score regression approach (similar to MTAG) or a test we included in the software).

We also observe similar results when training sfFDR on the various SNP selection strategies (detailed in comment #4; Figure S20).

We included these new results in the Results section:

To examine whether using uninformative data leads to false discoveries, we used the obesity-related traits from the UK Biobank study to assess sfFDR in a few "null" scenarios: (i) the trait values of BMI were permuted and three random chromosomes were replaced with the original BMI summary statistics, (ii) the trait values in the informative study were permuted to generate traits that were uncorrelated with BMI, and (iii) the trait values of BMI were permuted to remove any genetic signal. In scenario (i), the p_f -values of the null SNPs were well-behaved while MTAG p -values were substantially inflated when conditioning on BFP (Figure 3d; see Figure S16 for corresponding $\pi_0(z)$ and $f(p|z)$ estimates). In scenario (ii), the uninformative traits did not systematically inflate the significance of the p_f -values or MTAG's p -values (Figure S17, S16). In scenario (iii), there was a substantial inflation of MTAG's p -values when conditioning on BFP and a very slight inflation from the p_f -values (Figure S18, S16). Notably, MTAG attempts to stop the software when there is no evidence of a genetic signal in a trait (i.e., scenarios (ii) and (iii)) due to the potential of unstable estimates. Even though the slight inflation did not lead to any newly discovered SNPs in sfFDR, we investigated this setting

using simulated data and found that the bias arises from overfitting and can be minimized by adjusting the smoothing parameter in the density estimator and setting the proportion of truly null tests as constant (Figure S19). Finally, the observations from scenario (i)-(iii) hold regardless of the SNP selection strategy used in sfFDR (Figure S20).

and Methods section:

There were a few different “null” simulation settings considered. We implemented a scenario where the informative traits were permuted to be uncorrelated with BMI. In total, there were 10 permuted null datasets analyzed. We then considered another setting where we permuted BMI (10 times) and replaced 3 randomly selected chromosomes with the original BMI summary statistics. We designed the simulation this way so that we knew which SNPs were non-null while maintaining the genetic correlation across traits at the three selected chromosomes. Finally, we used the permuted null BMI summary statistics and the original (non-null) summary statistics of the informative traits. In this case, there is no genetic signal in the BMI summary statistics and so the proportion of truly null tests is 1.

6. Following up on comment #5, the corresponding $\pi_0(z)$ and $f(p|z)$ values from both the original analysis and the suggested analysis should be discussed. This would help explain the results and provide further insights into the robustness of the method.

We have added a new figure, Figure S16, to show the estimated $\pi_0(z)$ and $f(p|z)$ in the UK Biobank simulations (only using BFP for simplicity). In the original UKBB analysis, we observe that $\pi_0(z)$ varies as a function of BFP when the downsampling proportion is 1. When we condition on a null trait, we also observe that $\pi_0(z)$ is approximately constant and the functional density is highly correlated to the density estimates of the marginal distribution of the p-values. In the global null setting $\pi_0(z) = 1$, there is no relationship between BMI and BFP while the functional density estimates are centered

around 1 (which it should be as $f(p|z) = f(p) = 1$ in this setting). Finally, in the mixture of null and non-null chromosomes setting, there is a slightly more complex relationship: due to the underlying signal, there exists a non-constant $\pi_0(z)$ for the null SNPs (i.e., the proportion of truly tests varies as a function of the significance threshold because there are non-null SNPs).

We added the figure when discussing the null simulations in the UK Biobank section (see comment #5).

7. Unlike MTAG, the sfFDR framework does not output SNP-level effect sizes, which may limit its application in certain post-GWAS analyses. However, I do not think this limitation largely detracts from the novelty of the paper, provided the above comments are adequately addressed. Additionally, I appreciate the paper's focus on FDR-related work, though I remain concerned about how the LD structure might affect performance.

We note that our procedure also provides the posterior probabilities that a SNP is null which can be used for downstream analysis such as functional fine mapping (Figure S24), colocalization, and calculating the posterior mean of effect sizes. For example, given a set of estimated effect sizes, $\hat{\beta}_i$, and standard errors, \hat{s}_i , for SNPs $i = 1, 2, \dots, m$, we can calculate the posterior mean of the effect size β_i as

$$E[\beta_i | \hat{\beta}_i, \hat{s}_i, \mathbf{z}_i] = (1 - \Lambda_i(\mathbf{z}_i)) \hat{\beta}_i, \quad (3)$$

where Λ_i is the local FDR. We are exploring this as a future research direction to calculate polygenic risk scores while leveraging informative data.

Reviewer 2

In this manuscript, Bass et al. present sfFDR that leverages information from related traits to boost statistical power in genome-wide association studies (GWAS). The authors demonstrate the method's effectiveness through simulations and applications to both common traits and

a rare disease study, showing substantial gains in power equivalent to that from increasing sample size. The method and results appear to represent a useful contribution to the field. However, several major issues need to be addressed before the manuscript is suitable for publication.

Comments

1. In the main text (line 108-109), please specify what values “High”, “Medium” and “Low” represents and why these values are chosen. Currently, these values are only mentioned on line 532 and no justifications are made on why they are chosen as such. Similarly, on line 110, please specify why 1.25% and 2.50% are chosen as the number of overlapped non-null SNPs.

While the authors analyze a rare disease (EGPA) in their real data application, the simulations don't explicitly address rare disease scenarios. The authors should either clarify if the 'Low' scenario is meant to represent rare diseases, or add simulations specifically modeling rare disease characteristics (e.g., low prevalence, potentially larger effect sizes).

The prior probability of any SNP being null in our simulations is 0.98 which reflects the estimated value in the EGPA study (0.982). We used this value as the baseline prior and varied a few different parameters to study the performance of sfFDR under different settings. In particular, we changed the power of the primary study as “High”, “Medium” and “Low.” We selected the parameters in these setting to reflect a noticeably different number of discoveries at a genome-wide significance threshold. When the informative traits do not impact the primary study, the average number of discoveries is 1.31, 7.2, and 33.44 for the “Low”, “Medium” and “High” settings. The original EGPA identified two lead SNPs (ignoring MHC) and so we view the 'Low' setting as the closest simulation, while the 'High' setting reflects a study with a large sample size. We then varied the power of informative studies assuming that the power is substantially larger. We included a QQ-plot of a single realization in Figure S1 to help readers visualize the difference in choices, along with showing the LD-independent p -values from the EGPA study and informative

studies.

We also varied percentage of SNPs where the prior probability of a SNP being null is impacted by the informative traits. We randomly sampled values from 1.25% and 2.50% SNPs for each trait to represent a sparse overlap between the primary and informative trait, which makes the conservative assumption that the informative traits only impact a small percentage of SNPs. In a rarer disease GWAS, we would expect the study to be powered enough to identify a relationship between the primary and informative traits with a small percentage of non-null SNPs. Finally, the effect size strength (None, Moderate, Large) of the informative studies impact on the small subset of traits was varied to evaluate the strength of the relationship.

We added additional details in the Results section to clarify the parameter choices in the simulations:

We simulated the p -values for 150,000 independent SNPs in a primary study and three informative studies. The baseline prior probability of a null SNP was 0.98, which reflects our rare disease application (Figure S1). The signal strength of the studies (i.e., statistical power) was varied as “High,” “Medium,” and “Low.” As rarer disease studies usually only have enough power to identify a small number of non-null SNPs, we assumed that the informative traits influenced the prior probabilities of a small percentage of SNPs. In particular, the informative studies overlapped (shared non-null SNPs with the primary study) with randomly chosen values between 1.25% and 2.50% of the total number of SNPs to represent a sparse overlap with the primary trait. Finally, at the overlapping tests, the informative studies impacted both the prior probability of a SNP being null and the alternative density of the p -values with an effect size strength of “Large,” “Moderate,” and “None.”

2. The simulation should consider more realistic LD scenarios. Currently, the analysis uses UK Biobank subsets, but in practice, primary and informative GWASs often come from different cohorts with slightly different LD patterns. The authors should simulate these

cross-cohort LD differences to evaluate method robustness.

The UK Biobank analysis is designed to represent an ideal scenario where the LD is the same to evaluate the replication rate of discoveries made by sfFDR. As the reviewer mentions, the informative studies utilized in practice will likely have slightly different LD patterns. In order to evaluate this on real data while evaluating the replication rate, we used summary statistics of BMI from FinnGen [7], GIANT [8], and the Millions Veterans Program (MVP) [9]. When leveraging summary statistics from either of these cohorts, our main conclusions remain similar which suggests that sfFDR is robust to slight LD differences.

We added these new findings in the Results section:

We also evaluated the robustness of sfFDR when leveraging summary statistics from other European populations with different LD patterns. In particular, we used the BMI summary statistics from FinnGen [7], GIANT [8], and the Millions Veterans Program (MVP) [9]. When using BMI from the other biobanks as an informative trait, sfFDR and MTAG substantially increased the number of discoveries compared to the standard analysis and nearly all were replicated by a meta-analysis (Figure S15). We found that MTAG outperformed sfFDR at lower samples sizes while sfFDR outperformed MTAG at larger sample sizes. We note that this is an ideal application of MTAG since the model assumes that the covariance matrix of true SNP effects is the same across all SNPs (i.e., the genetic architecture between the primary and informative traits are identical). These results suggest that sfFDR is robust to slight LD differences across the primary and informative traits.

And in the Discussion section:

Finally, while our results suggest that sfFDR is robust to slight LD differences across the primary and informative traits, we do not recommend using summary statistics from populations that are very different (in terms of ancestry) than the primary study.

3. In the simulation settings, the authors should also consider the scenarios when the causal variants are not included by the genotyped SNPs. Often in real-world scenarios, the causal variant is untyped and the genotyped SNPs are partially tagging it. The authors should investigate (or at least comment on) how their method would behave under such circumstances.

As the reviewer points out, whether a causal SNP is measured or tagged can impact the performance of sfFDR and the standard GWAS analysis approach. For a typed causal SNP, we would expect more power at that location while, for a tagged causal SNP, there will be a power loss. The power loss is related to how close the SNP is to the causal SNP (i.e., how well the SNP tags the causal SNP). Because the underlying status is unknown in real data, this requires a flexible definition of a 'true' positive as a SNP that is either a causal SNP or tags a causal SNP. We note that this assumption is implicitly made by standard GWAS approaches. Finally, we can view our simulation study as the causal SNP setting and the UK Biobank study as the tagged SNP setting. We found that sfFDR's functional p -value provided type I error rate control in both settings.

We clarified this in our discussion:

Fourth, since it is not possible to distinguish whether a (tagged) SNP is a true discovery or is capturing a nearby causal SNP due to LD, we defined a true discovery as a SNP that either tags or is the causal SNP. Our results showed that the functional p -value controls the type I error rate when either the causal variant is measured (as in the simulation study) or tagged (as in the UK Biobank analysis).

4. In Supplementary Figure 8, while the median empirical FDRs at level 0.01 are approximately 0.01, all methods show long tails in their distributions. Does this suggest poorer FDR calibration in the presence of LD? The authors should discuss the implications for practical applications.

It is important to first note that the FDR is a quantity that is controlled in expectation. While there was variation in individual simulation studies around the true value, sfFDR controlled the FDR in expectation. Furthermore, our simulations represent an extreme case of LD where the primary and informative study p -values are the same within each LD block, which will substantially increase the observed variability of the proportion of false discoveries compared to real data. Our goal was to illustrate that the functional p/q -values from sfFDR were controlling the type I error rate and FDR in expectation in such an extreme setting. While the FDR is controlled in expectation, the variance of the proportion of false discoveries is larger in the presence of LD. We note that a similar increase in variability is also observed with the type I error rate. These observations are not sfFDR-specific but are also observed in the q -values and p -values of dependent data.

We added to the Results section to include the above insights:

Finally, we assessed control of the type I error rate and FDR in the dependent SNP setting. We first randomly assigned each independent SNP an LD block size based on the empirical distribution from the UK Biobank (Methods). Given the block size, we then duplicated the p -values for the primary and informative studies so that the LD block was perfectly correlated. While this represents an unrealistic scenario, it is a deliberately challenging setting to evaluate estimates in the sfFDR framework because perfect correlations will inflate variability of the type I error rate and FDR compared to real data. Even under such an extreme case, we found that the estimated p_f -value and q_f -value from sfFDR controlled the type I error rate and FDR in expectation, respectively (Figure S10, S11, S12). As expected, due to the dependence from LD, the variability of the type I error rate and FDR was larger compared to the independent SNP setting. Nevertheless, the estimated p_f -value and q_f -value had similar variability to the standard p -value and q -value, respectively.

5. In the benchmark analyses, the authors should include comparisons with established summary-level multi-trait methods, particularly MTAG (Turley et al., Nature Genetics

2018) and similar approaches. While these methods use different statistical frameworks, they share the core goal of leveraging summary statistics from related traits to enhance GWAS power. A systematic comparison would help users understand when to prefer sfFDR over existing methods.

We thank the reviewer for this suggestion. Another reviewer had the same recommendation and so we added comprehensive simulations comparing sfFDR to MTAG: Please see our detailed response in comments #1 and #5 of Reviewer 1.

To briefly summarize our results, MTAG makes a strict assumption regarding the correlation between effect sizes across traits which can lead to false discoveries. We demonstrate that it can substantially inflate the type I error rate in the UKBB study, suggesting that MTAG may not be reliable when the informative traits have different genetic architectures to the primary trait of interest. We believe that this result distinguishes sfFDR from MTAG and will help increase the general interest and usefulness of sfFDR.

6. In the BMI analysis, the authors only tested scenarios where conditioning traits are either all informative or all uninformative. However, in real practice, users might have a mix of both types when selecting traits. The authors should evaluate how sfFDR performs with a mixture of informative and uninformative conditioning traits, particularly how the proportion of uninformative traits affects power. This would provide practical guidance for users who may not be able to perfectly select relevant traits.

We have updated the manuscript to address the reviewers concern. Specifically, we used the set of permuted null UK Biobank traits and the corresponding set of non-null UK Biobank traits. We then fit sfFDR to a single non-null trait while changing the number of null trait traits (so 1/2, 1/3, 1/4 of the informative traits are null). We then repeated this process by adding the second (and third) non-null traits while varying the number null traits. We found that the proportion of null traits did not have a substantial impact on the results in the UK Biobank study. As such, selecting potentially null traits did not result in poor performance.

We added these simulations in the Results section:

We then evaluated sfFDR in a setting where the conditioning traits were a mixture of uninformative (the permuted null traits from scenario (ii)) and informative (Figure S21). To do so, we fit sfFDR to a mixture of one, two, or three informative and uninformative traits. We found that the added uninformative traits did not have a substantial impact on the number of discoveries and nearly all of the discoveries were replicated with the meta-analysis approach. Therefore, including potentially non-informative traits did not result in poor performance in sfFDR.

We do note that sfFDR will run into issues when there are a substantial number of traits (see Discussion). In particular, the logistic regression model implemented will run into known fitting issues as the number of traits becomes large. In such cases regularization (e.g., LASSO) can be used to improve the fit in such cases. While we did not run into issues in this work, it may be an issue when fitting a substantial number of informative variables (such as functional annotations), and we are planning to extend the model fitting in such applications in future work.

7. For the novel EGPA signals identified by sfFDR, validation is crucial. The authors should either attempt replication in independent EGPA cohorts, or demonstrate through simulations (relevant to point 2 above). This is particularly important given sfFDR's potential utility for rare disease studies.

We agree with the reviewer that the gold standard in any GWAS is validation through a discovery data set. However, in the EGPA study, nearly all of the accessible EGPA patients in Europe were used in the study (annual incidence of 1-2 cases per million [10]). In fact, this is a common challenge for many rare disease studies: a majority of the available samples are used to maximize power and any validation set is often too underpowered.

As such, we address the Reviewers concern by adding new analyses to diseases with

a low number of cases (expected in rare disease studies) where independent validation data are available:

The sfFDR framework offers potential benefits in the rare disease setting because it is difficult and costly to acquire additional samples to improve power. Therefore, we first examined sfFDR in studies with a small number of cases where independent validation data was available. In particular, we used four diseases in the FinnGen biobank, namely, autoimmune thyroiditis (ATH; 688 cases and 424,208 controls), juvenile idiopathic arthritis (JIA; 788 cases and 172,834 controls), myositis (MYO; 932 cases and 357,549 controls), and systemic lupus erythematosus (SLE; 835 cases and 232,612 controls; Methods). In total, the standard analysis identified 4 lead SNPs (1 in ATH and 3 in SLE) at the genome-wide significance threshold (Table S1). Leveraging summary statistics from disease-relevant traits (rheumatoid arthritis [11] and hypothyroidism [12]), sfFDR identified 8 additional lead SNPs where the p -values of 6 were genome-wide significant and 2 were below 5×10^{-7} in the independent studies. Thus, we found that the discoveries from sfFDR were replicated in the rare disease setting, in line with our UK Biobank and simulation studies.

More generally, we designed the UK Biobank study so we could validate any new discoveries from sfFDR with an independent (left out) data set. Furthermore, as noted in comment #1, we also designed the simulations to reflect the sparsity level observed in the EGPA study to evaluate the performance as a function of study power of the primary and informative traits. We then evaluated the genetic overlap of the new discoveries in the EGPA study with the clinically relevant traits and, more generally, as relevant immune-related discoveries for a follow-up analysis.

8. More concrete guidelines for selecting informative traits would be valuable, particularly for rare diseases. Can metrics like genetic correlation estimates guide trait selection? The authors should also clarify how they define 'high-powered' traits (line 293) - is this based on significant variant count, heritability estimates, or other metrics?

Genetic correlation estimates can help guide choices in trait selection, although we caution against a “shotgun” approach where the largest correlation among hundreds of traits is chosen. This will lead to selection bias issues. We are working on providing a LASSO implementation to handle a massive amount of traits to avoid such selection bias problems. Given the plethora of summary statistics of thousands of traits across multiple biobanks, a safer approach is to identify co-morbidities or shared traits among the population of interest (e.g., from expertise or using previous research). We show that sfFDR is robust to non-informative traits and so a practitioner can integrate multiple sets of GWAS summary statistics.

We also clarified the use of the vague term ‘high-powered’: In agreement with the reviewer, we note that the best informative traits depends on the underlying genetic architecture and relationship between traits, but a guideline is to try to select studies with large sample sizes so sfFDR can adaptively leverage the trait information to improve power. We note that we studied the impact of informative trait study power in the simulation study.

We clarified how to select informative traits in the Discussion section:

Our results have implications for the design of pleiotropy-informed significance analyses. As expected, the power improvements with sfFDR increased whenever the study power increased, for both the primary and informative studies. As such, practitioners should use informative traits with large sample sizes in sfFDR. Fortunately, there is a large collection of GWAS summary statistics in publicly available repositories for thousands of complex traits (see, e.g., [13, 14]), although selecting the informative traits *a priori* will require careful consideration to avoid model selection (and fitting) problems. Identifying polygenic traits that have a similar genetic architecture with the primary trait will provide the largest increase in power, and so traits that are too dissimilar (or with little polygenicity) will not be ideal informative traits. In our rare disease analysis, we chose informative traits based on known co-morbidities among the patients. Another strategy is to select informative traits based on the esti-

mated genetic correlation with the primary trait. However, we caution against a “shotgun” approach where the largest correlations among hundreds of traits are chosen as this can lead to selection bias. Importantly, our results showed that sfFDR is robust to the inclusion of non-informative traits. Thus, sfFDR can integrate multiple sets of candidate informative traits with different genetic architectures. In contrast, MTAG did not improve upon the standard analysis when conditioning on the three obesity-related traits and can also inflate the type I error rate. While our method can incorporate many informative traits, handling a very large number of informative traits may require dimensionality reduction (e.g., principal component analysis [15] or sliced inverse regression [16]), variable selection, or regularization to reduce the computational time and allow for stable model fitting in sfFDR (Figure S14).

9. The authors should discuss how different genetic architectures (e.g., polygenicity levels, effect size distributions across allele frequencies) might impact sfFDR performance, even if the method is theoretically robust to these factors.

In general, we anticipate that leveraging traits with similar genetic architectures will provide the most increase in power while, as the dissimilarity between traits increases, the smaller the power improvements will be. Furthermore, primary and informative traits that are more polygenic will provide a larger set of non-null SNPs and thus provide more information for sfFDR to use in its procedure. In this case, the functional relationship will be easier to learn which can lead to larger improvements in power. On the extreme end, an informative trait with very little polygenicity will not be useful as the underlying signal is too sparse to be leveraged within our framework. In general, we expect sfFDR to be best suited for primary/informative traits that are polygenic where the higher the polygenicity the better we anticipate sfFDR will perform.

We added these details in the Discussion (see response to comment #8 for added text).

10. The authors should report runtime and memory requirements for typical GWAS sum-

mary statistics with different number of variants and/or conditioning GWASs, ideally in comparison to other existing methods.

We added a new figure to show the computational time as a function of the number of informative traits and LD-independent SNPs trained using a single core of an Apple M3 processor (Figure S14).

We reference the new figure in a couple places in the text. In the Results section,

Since our primary application is rare diseases, we selected SNPs based on haplotype blocks with informative trait sampling to maximize coverage of pleiotropic SNPs and minimize computation time (Figure S14).

and in the Discussion section,

While our method can incorporate many informative traits, handling a very large number of informative traits may require dimensionality reduction (e.g., principal component analysis [15] or sliced inverse regression [16]), variable selection, or regularization to reduce the computational time and allow for stable model fitting in sfFDR (Figure S14).

Reviewer 3

This manuscript proposes a novel method to exploit currently published GWAS summary statistics related to the main trait under investigation. By using the surrogate functional false discovery rate (sfFDR), the authors show that their method increases power to discover new loci associated to the main trait by leveraging the summary statistics coming from other related traits. Furthermore, the method outputs a functional p-value and Bayes factor that could help fine-mapping and colocalization. Simulations show that their method appropriately controls for FDR and Type 1 error while increasing power over standard p-values. Also, one of the strengths of their method is that it allows either the use of the FDR (more popular in eQTL studies) or FWER (more popular in GWAS) control since there is a direct mapping between

the functional q-values and functional p-values.

The manuscript is well-written, easy to follow and the Methods section not too difficult to read even for a non-statistics audience. The Abstract, Introduction and Discussion sections are clear. The reference list is complete and appropriate. I only have a few comments:

1. I found a somewhat “strange” behavior in Figure 2b and Supp Fig 6. When the effect size of informative studies is “Moderate,” the number of discoveries is always lower than when the effect size is “None.” Could the authors explain why?

We thank the reviewer for this observation. We had erroneously set the proportion of truly null tests being 0.95 in the “None” setting while it should be 0.98 to match the other settings. The figures have been updated and the number of discoveries increases as the prior strength increases as expected.

2. Is there any prior studies or literature suggesting that the overlap of shared non-null SNPs (i.e. pleiotropy) between informative studies and the primary study is around 1.25% or 2.50%? Can it be modified by the user when using the software?

The primary purpose of the simulation study was to demonstrate the method in a sparse setting likely encountered in the rarer disease setting. As such, the selection of these values is only relevant for how we simulated the data, and sfFDR makes no such assumption about the overlap percentage. In particular, due to small sample sizes, the estimated proportion of truly null tests is high in the absence of any information for rare disease studies (in our simulations we assigned this to be 0.98). For example, in the EGPA study, the estimated proportion of truly null tests is .982 (see Figure S1 for a comparison of the simulation p-values for the primary and informative traits compared to the EGPA study). Given the limited sample sizes (and power) of such studies, we reasoned that a genetically correlated trait will only be powered enough to impact the prior of a small percentage of SNPs (1.25% or 2.50%).

The only necessary choice by the user in our software is the location of the knots in the natural cubic spline, and we have a function that helps the user create the model

(`pi0_model` function in the software). Ideally, these knots should be chosen at the small quantiles of the information traits p -values where the non-null SNPs are likely to be located. The `sffdr` package then models the relationship for the user.

We clarified our simulation study in the Results section:

We simulated the p -values for 150,000 independent SNPs in a primary study and three informative studies. The baseline prior probability of a null SNP was 0.98, which reflects our rare disease application (Figure S1). The signal strength of the studies (i.e., statistical power) was varied as “High,” “Medium,” and “Low.” As rarer disease studies usually only have enough power to identify a small number of non-null SNPs, we assumed that the informative traits influenced the prior probabilities of a small percentage of SNPs. In particular, the informative studies overlapped (shared non-null SNPs with the primary study) with randomly chosen values between 1.25% and 2.50% of the total number of SNPs to represent a sparse overlap with the primary trait. Finally, at the overlapping tests, the informative studies impacted both the prior probability of a SNP being null and the alternative density of the p -values with an effect size strength of “Large,” “Moderate,” and “None.”

3. Somehow related to point 2 above, is there any recommendation as to the number of informative studies the user should include before running the method? Does this choice be guided by moderate-to-strong genetic correlations between the traits?

We believe that this choice should be guided by expertise to identify clinically relevant traits which one would expect to be moderately-to-strongly correlated with the primary traits of interest. As `sffdr` can handle many traits, the more informative traits included, the higher the power improvements will be. Additionally, we showed that including uninformative traits does not impact the type I error rate.

For the case where there are a substantial number of informative traits (more likely in the common disease setting), we will provide users with a LASSO or ridge regression option

to handle such cases. While another effective strategy can be based on calculating the genetic correlation between the primary trait and informative traits, we urge caution with such an approach as selection bias may occur.

We updated our Discussion to further address this question:

Our results have implications for the design of pleiotropy-informed significance analyses. As expected, the power improvements with sfFDR increased whenever the study power increased, for both the primary and informative studies. As such, practitioners should use informative traits with large sample sizes in sfFDR. Fortunately, there is a large collection of GWAS summary statistics in publicly available repositories for thousands of complex traits (see, e.g., [13, 14]), although selecting the informative traits *a priori* will require careful consideration to avoid model selection (and fitting) problems. Identifying polygenic traits that have a similar genetic architecture with the primary trait will provide the largest increase in power, and so traits that are too dissimilar (or with little polygenicity) will not be ideal informative traits. In our rare disease analysis, we chose informative traits based on known co-morbidities among the patients. Another strategy is to select informative traits based on the estimated genetic correlation with the primary trait. However, we caution against a “shotgun” approach where the largest correlations among hundreds of traits are chosen as this can lead to selection bias. Importantly, our results showed that sfFDR is robust to the inclusion of non-informative traits. Thus, sfFDR can integrate multiple sets of candidate informative traits with different genetic architectures. In contrast, MTAG did not improve upon the standard analysis when conditioning on the three obesity-related traits and can also inflate the type I error rate. While our method can incorporate many informative traits, handling a very large number of informative traits may require dimensionality reduction (e.g., principal component analysis [15] or sliced inverse regression [16]), variable selection, or regularization to reduce the computational time and allow for stable model fitting in sfFDR (Figure S14).

4. Small typos:

- Line 201: "... of EGPAS is unknown ..."
- Line 226: " ... there were 226 discoveries ..."

We thank the reviewer for finding these typos and have corrected them in the manuscript. We note that there were 226 discoveries in our analysis of EGPA.

5. In summary, this novel method will be of interest to many practitioners in the GWAS community. I tried to install the package but the "qvalue" package was not available in the latest version of R (4.4.2):

"Warning in install.packages : package 'qvalue' is not available for this version of R"

The qvalue package needs to be updated first before running sffdr.

We found that this issue is because CRAN does not install the qvalue package (it is located on Bioconductor). We updated our installation instructions on GitHub to

```
# Install q-value package
if (!require("BiocManager", quietly = TRUE))
  install.packages("BiocManager")
```

```
BiocManager::install("qvalue")
```

```
# Install sffdr
install.packages("sffdr")
```

Running the above code should fix this issue.

1 Main figures and tables

Figure 1: Overview of the surrogate functional false discovery rate (sfFDR) framework. **(a)** Estimate the functional local FDR of the primary GWAS p -values given a set of informative summary statistics. The functional local FDR values are used for **(b)** estimating the functional q -value (q_f -value) and functional p -value (p_f -value) to control the FDR and family-wise error rate, respectively, and **(c)** functional fine mapping.

Figure 2: Simulation results for the sfFDR framework in the independent SNP simulation study when the primary study power is “Medium.” **(a)** The average number of discoveries as a function of the target false discovery rate (FDR) using the standard q -value (dark orange), functional q -value from sfFDR (green), and oracle functional q -value (black). **(b)** The number of discoveries using the standard p -value (grey), functional p -value from sfFDR (blue), and oracle functional p -value (black) at a genome-wide significance threshold of 5×10^{-8} . We varied the power of the informative studies (columns) and the effect size strength of the informative studies (top plot: shape; bottom plot: x-axis). There were a total of 500 replicates at each setting.

Figure 4: Significance results for the EGPA study. (a) The prior probability of a test being null as a function of the surrogate variable; (b) The functional p -value from sfFDR versus the standard p -value of the study; (c) The number of significant tests at various p -value thresholds for the functional and standard p -values. (d) The functional q -value versus functional p -value relationship. The above plot shows SNPs with standard p -values below 1×10^{-4} in (a)-(b) and functional p -values below 5×10^{-8} in (c)-(d).

Figure 5: Manhattan plot of the (a) standard p -values and the (b) functional p -values (p_f -values) from sfFDR in the EGPA study. The red line represents the genome-wide significance threshold of 5×10^{-8} . The lead SNPs were assigned to the nearest genes. Note that p -values below 0.05 are removed from the plot.

Chr	rsid	Gene	MAF	Informative traits										P_f	Q_f
				EGPA		ASTAO		ASTCO		EOSC					
				β	P	β	P	β	P	β	P				
5	rs1837253:C>T	TSLP	0.258	-0.41	7.96×10^{-10}	-0.08	1.50×10^{-17}	-0.17	5.50×10^{-37}	-0.04	1.89×10^{-22}	5.35×10^{-14}	4.27×10^{-7}		
2	rs144569746:T>C	BCL2L11	0.107	-0.57	1.54×10^{-9}	-0.06	1.70×10^{-5}	-0.06	2.80×10^{-3}	-0.06	2.57×10^{-26}	1.51×10^{-12}	6.01×10^{-6}		
5	rs10066308:A>G	IRF1	0.305	-0.35	6.95×10^{-8}	-0.07	2.30×10^{-13}	-0.09	5.10×10^{-13}	-0.04	8.18×10^{-32}	1.33×10^{-11}	1.18×10^{-5}		
10	rs7898135:A>C	GATA3	0.283	0.31	2.72×10^{-6}	0.10	1.20×10^{-26}	0.10	1.50×10^{-14}	0.04	6.97×10^{-23}	2.01×10^{-10}	5.74×10^{-5}		
6	rs11754356:T>C	BACH2	0.394	0.27	7.14×10^{-6}	0.05	2.70×10^{-10}	0.09	1.10×10^{-13}	0.03	4.80×10^{-19}	4.90×10^{-10}	1.12×10^{-4}		
21	rs8133843:A>G	RUNX1	0.373	-0.30	9.69×10^{-7}	-0.04	4.40×10^{-5}	-0.02	3.90×10^{-2}	-0.03	7.90×10^{-12}	2.95×10^{-9}	3.51×10^{-4}		
3	rs9825301:T>G	TPRG1	0.314	-0.29	4.05×10^{-6}	-0.03	1.60×10^{-4}	-0.05	1.30×10^{-5}	-0.03	1.51×10^{-14}	6.82×10^{-9}	5.73×10^{-4}		
17	rs12952581:A>G	ZNF652	0.143	-0.24	7.96×10^{-5}	-0.05	1.90×10^{-7}	-0.09	8.40×10^{-14}	-0.03	2.67×10^{-13}	3.29×10^{-8}	1.39×10^{-3}		
12	rs10876864:A>G	IKZF4	0.416	0.23	1.19×10^{-4}	0.06	1.40×10^{-12}	0.10	1.10×10^{-17}	0.03	6.24×10^{-13}	3.61×10^{-8}	1.47×10^{-3}		
11	rs7927997:T>C	LRRC32	0.395	-0.22	2.37×10^{-4}	-0.08	5.20×10^{-19}	-0.17	6.20×10^{-46}	-0.04	7.32×10^{-27}	3.86×10^{-8}	1.53×10^{-3}		

Table 1: Functional p -values (P_f) and q -values (Q_f) of the lead SNPs from the EGPA analysis. The SNP identifiers are given as rsid:reference_allele>effect_allele. The informative traits were adult-onset asthma (ASTAO), childhood-onset asthma (ASTCO), and eosinophil count (EOSC).

References

- [1] Turley P, Walters RK, Maghzian O, Okbay A, Lee JJ, Fontana MA, et al. Multi-trait analysis of genome-wide association summary statistics using MTAG. *Nature Genetics*. 2018;50(2):229–237. doi:10.1038/s41588-017-0009-4.
- [2] Dahl A, Thompson M, An U, Krebs M, Appadurai V, Border R, et al. Phenotype integration improves power and preserves specificity in biobank-based genetic studies of major depressive disorder. *Nature Genetics*. 2023;55(12):2082–2093. doi:10.1038/s41588-023-01559-9.
- [3] Lei L, Fithian W. AdaPT: an interactive procedure for multiple testing with side information. *Journal of the Royal Statistical Society Series B: Statistical Methodology*. 2018;80(4):649–679. doi:10.1111/rssb.12274.
- [4] Zhang X, Chen J. Covariate adaptive false discovery rate control with applications to omics-wide multiple testing. *Journal of the American Statistical Association*. 2022;117(537):411–427. doi:10.1080/01621459.2020.1783273.
- [5] Boca SM, Leek JT. A direct approach to estimating false discovery rates conditional on covariates. *PeerJ*. 2018;6:e6035.
- [6] Speed D, Hemani G, Johnson MR, Balding DJ. Improved heritability estimation from genome-wide SNPs. *The American Journal of Human Genetics*. 2012;91(6):1011–1021. doi:10.1016/j.ajhg.2012.10.010.
- [7] Kurki MI, Karjalainen J, Palta P, Sipilä TP, Kristiansson K, Donner KM, et al. FinnGen provides genetic insights from a well-phenotyped isolated population. *Nature*. 2023;613(7944):508–518. doi:10.1038/s41586-022-05473-8.
- [8] Locke AE, Kahali B, Berndt SI, Justice AE, Pers TH, Day FR, et al. Genetic studies of body mass index yield new insights for obesity biology. *Nature*. 2015;518(7538):197–206. doi:10.1038/nature14177.

- [9] Verma A, Huffman JE, Rodriguez A, Conery M, Liu M, Ho YL, et al. Diversity and scale: Genetic architecture of 2068 traits in the VA Million Veteran Program. *Science*. 2024;385(6706):eadj1182. doi:10.1126/science.adj1182.
- [10] Lyons PA, Peters JE, Alberici F, Liley J, Coulson RMR, Astle W, et al. Genome-wide association study of eosinophilic granulomatosis with polyangiitis reveals genomic loci stratified by ANCA status. *Nature Communications*. 2019;10(1):5120. doi:10.1038/s41467-019-12515-9.
- [11] Okada Y, Wu D, Trynka G, Raj T, Terao C, Ikari K, et al. Genetics of rheumatoid arthritis contributes to biology and drug discovery. *Nature*. 2014;506(7488):376–381. doi:10.1038/nature12873.
- [12] Mbatchou J, Barnard L, Backman J, Marcketta A, Kosmicki JA, Ziyatdinov A, et al. Computationally efficient whole-genome regression for quantitative and binary traits. *Nature Genetics*. 2021;53(7):1097–1103. doi:10.1038/s41588-021-00870-7.
- [13] MacArthur J, Bowler E, Cerezo M, Gil L, Hall P, Hastings E, et al. The new NHGRI-EBI Catalog of published genome-wide association studies (GWAS Catalog). *Nucleic acids research*. 2017;45(D1):D896–D901.
- [14] Watanabe K, Stringer S, Frei O, UmićevićMirkov M, de Leeuw C, Polderman TJC, et al. A global overview of pleiotropy and genetic architecture in complex traits. *Nature Genetics*. 2019;51(9):1339–1348. doi:10.1038/s41588-019-0481-0.
- [15] Jolliffe IT. *Principal Component Analysis*. Springer Series in Statistics. Springer; 2002.
- [16] Li KC. Sliced Inverse Regression for dimension reduction. *Journal of the American Statistical Association*. 1991;86(414):316–327.
- [17] Sakaue S, Kanai M, Tanigawa Y, Karjalainen J, Kurki M, Koshihara S, et al. A cross-population atlas of genetic associations for 220 human phenotypes. *Nature Genetics*. 2021;53(10):1415–1424. doi:10.1038/s41588-021-00931-x.
- [18] López-Isac E, Smith SL, Marion MC, Wood A, Sudman M, Yarwood A, et al. Combined genetic analysis of juvenile idiopathic arthritis clinical subtypes identifies novel risk

loci, target genes and key regulatory mechanisms. *Annals of the Rheumatic Diseases*. 2021;80(3):321–328. doi:10.1136/annrheumdis-2020-218481.

[19] Rothwell S, Amos CI, Miller FW, Rider LG, Lundberg IE, Gregersen PK, et al. Identification of Novel Associations and Localization of Signals in Idiopathic Inflammatory Myopathies Using Genome-Wide Imputation. *Arthritis & Rheumatology*. 2023;75(6):1021–1027. doi:<https://doi.org/10.1002/art.42434>.

[20] Bentham J, Morris DL, Cunninghame Graham DS, Pinder CL, Tombleson P, Behrens TW, et al. Genetic association analyses implicate aberrant regulation of innate and adaptive immunity genes in the pathogenesis of systemic lupus erythematosus. *Nature Genetics*. 2015;47(12):1457–1464. doi:10.1038/ng.3434.

2 Supplementary materials

2.1 Supplementary tables

Disease	Chr	rsid	Gene	MAF	β	P	Informative traits				P_f	Discovery P
							Rheumatoid Arthritis		Hypothyroidism			
							β	P	β	P		
ATH (R12)	1	rs6679677:C>A	RSBN1	0.142	0.53	2.75×10^{-15}	0.59	3.10×10^{-149}	0.38	1.44×10^{-154}	1.18×10^{-19}	5.37×10^{-44} [17]
	12	rs7137828:C>T	ATXN2	0.410	-0.23	2.04×10^{-5}	-0.08	5.30×10^{-7}	-0.22	1.70×10^{-112}	3.29×10^{-9}	4.72×10^{-25} [17]
	2	rs3087243:G>A	CTLA4	0.333	-0.21	2.26×10^{-4}	-0.14	9.20×10^{-20}	-0.18	9.42×10^{-77}	5.71×10^{-9}	3.62×10^{-14} [17]
JIA (R5)	12	rs653178:C>T	ATXN2	0.470	0.05	1.83×10^{-5}	-0.08	5.60×10^{-7}	-0.21	1.07×10^{-111}	2.88×10^{-9}	1.82×10^{-9} [18]
	1	rs6679677:C>A	RSBN1	0.179	0.07	1.66×10^{-3}	0.59	3.10×10^{-149}	0.38	1.44×10^{-154}	1.81×10^{-8}	9.18×10^{-14} [18]
MYO (R12)	1	rs6679677:C>A	RSBN1	0.145	0.28	4.97×10^{-6}	0.59	3.10×10^{-149}	0.38	1.44×10^{-154}	1.03×10^{-8}	2.00×10^{-7} [19]
SLE (R7)	7	rs3778754:C>G	IRF5	0.429	0.34	9.09×10^{-13}	0.10	1.80×10^{-11}	0.05	8.03×10^{-7}	1.41×10^{-15}	1.59×10^{-36} [20]
	2	rs4274624:C>T	STAT4	0.227	-0.32	3.90×10^{-9}	-0.13	6.90×10^{-12}	-0.14	1.63×10^{-35}	4.02×10^{-12}	9.73×10^{-66} [20]
	1	rs17849501:G>T	NCF2,SMG7	0.038	0.70	1.05×10^{-11}	0.10	5.00×10^{-2}	0.07	7.93×10^{-4}	1.39×10^{-11}	1.81×10^{-59} [20]
	8	rs998683:G>A	BLK	0.260	0.28	1.96×10^{-7}	0.09	1.40×10^{-6}	-0.03	7.29×10^{-3}	9.49×10^{-10}	1.26×10^{-14} [20]
	1	rs6679677:C>A	RSBN1	0.140	0.27	3.43×10^{-5}	0.59	3.10×10^{-149}	0.38	1.44×10^{-154}	1.12×10^{-9}	4.54×10^{-13} [20]
	12	rs10774624:G>A	SH2B3	0.400	-0.20	4.48×10^{-5}	-0.08	2.40×10^{-7}	-0.21	1.78×10^{-101}	5.39×10^{-9}	1.49×10^{-7} [20]

Table S1: Functional p -values (P_f) and q -values (Q_f) of the lead SNPs when applying sfFDR to autoimmune thyroiditis (ATH; 688 cases and 424,208 controls), juvenile idiopathic arthritis (JIA; 788 cases and 172,834 controls), myositis (MYO; 932 cases and 357,549 controls), and systemic lupus erythematosus (SLE; 835 cases and 232,612 controls) in the FinnGen biobank [7]. The informative traits were rheumatoid arthritis [11] and hypothyroidism [12]. Note that the FinnGen release versions R5 (JIA), R7 (SLE) and R12 (MYO, ATH) were used and the SNP identifiers are given as rsid:reference_allele>effect_allele.

Gene	sfFDR	Standard
BACH2	66	167
BCL2L11	13	14
GATA3	112	272
IKZF4	45	1,113
IRF1	52	39
LRRC32	136	2,564
RUNX1	141	68
TPRG1	45	64
TSLP	1	1
ZNF652	100	1,093

Table S2: The size of the 95% credible set using sfFDR and the standard (or original) p -values in the EGPA study. Note that only *TSLP* and *BCL2L11* are below genome-wide significance level for the standard p -values.

Gene	ASTAO	ASTCO	EOSC
BACH2	0.196	0.205	0.153
BCL2L11	0.968	0.000	0.850
GATA3	0.254	0.058	0.226
IKZF4	0.194	0.343	0.791
IRF1	0.320	0.000	0.000
LRRC32	0.131	0.161	0.112
RUNX1	0.076	0.081	0.000
TPRG1	0.173	0.000	0.822
TSLP	0.000	0.963	0.963
ZNF652	0.567	0.258	0.439

Table S3: The proportion of SNPs in the 95% credible set from sfFDR that overlap with the credible sets from the informative studies (ASTAO, ASTCO, and EOSC).

2.2 Supplementary figures

Figure S1: A QQ-plot of a single realization from the independent SNP simulation study. The power of the (a) primary study and (b) the three informative studies (shown in black) were varied as “High,” “Medium,” and “Low.” We plotted the set of LD-independent SNPs from the EGPA (dark red), eosinophil count (EOSC; dark green), adult-onset asthma (ASTAO; dark orange), and childhood-onset asthma (ASTCO; dark blue) studies for comparison to real data benchmarks.

Figure S2: Assessing the target FDR at level 0.01 using the oracle functional q -values, standard q -values, functional q -values from sfFDR, Adapt, CAMT, and Boca-Leek in the independent SNP setting. The boxplot combines the “None,” “Moderate,” and “Large” effect size strength settings.

Figure S3: The number of discoveries as a function of the target false discovery rate (FDR) in the independent SNP simulation study using the standard q -value (dark orange), functional q -value from sfFDR (green), and the oracle functional q -value (black). We varied the power of the primary study (rows), the power of the informative studies (columns), and the effect size strength of the informative studies (shape). Each point is the average from 500 replicates.

Figure S4: Comparing the root mean square error (RMSE) of the estimated proportion of truly null tests from the standard q -value, sfFDR, Adapt, CAMT, and Boca-Leek in the independent SNP setting. There were a total of 500 replicates at each combination of primary study power (rows), informative study power (columns), and the effect size strength of the informative studies (color).

Figure S5: Comparing the estimated proportion of truly null tests from the standard q -value, sfFDR, Adapt, CAMT, and Boca-Leek in the independent SNP setting. There were a total of 500 replicates at each combination of primary study power (rows), informative study power (columns), and the effect size strength of the informative studies (color).

Figure S6: Comparing the root mean square error (RMSE) of the log-transformed estimated conditional density of the p -values given the informative traits using the standard q -value (i.e., the marginal density), sfFDR, Adapt, and CAMT in the independent SNP setting. There were a total of 500 replicates at each combination of primary study power (rows), informative study power (columns), and the effect size strength of the informative studies (color).

Figure S7: The empirical true discovery rate as a function of the empirical false discovery rate at a target FDR level of 0.001, 0.002, . . . , 0.01 in the independent SNP simulation study using the oracle functional q -value (black), functional q -value from sfFDR (green), CAMT (light blue), Boca-Leek (blue), AdaPT (orange), and standard q -value (dark orange). We varied the power of the primary study (rows), the power of the informative studies (columns), and the effect size strength of the informative studies (shape). Each point is the average from 500 replicates.

Figure S8: The type I error rate of the standard p -values, functional p -values from sfFDR, and the oracle functional p -values at a significance threshold of 1×10^{-4} in the independent SNP setting. We varied the power of the primary study (rows), the power of the informative studies (color), and the effect size strength of the informative studies (x-axis). There were a total of 500 simulations at each setting.

Figure S9: A comparison of the number of discoveries in the independent SNP simulation study using the standard p -value (grey), functional p -value from sfFDR (blue), and oracle functional p -value (black) at a significance threshold of 5×10^{-8} . We varied the power of the primary study (rows), the power of the informative studies (columns), and the effect size strength of the informative studies (x-axis). There were a total of 500 simulations at each setting.

Figure S10: The type I error rate of p -values, functional p -values, and the oracle functional p -values at a significance threshold of 1×10^{-4} in the dependent SNP setting. We varied the power of the primary study (rows), the power of the informative studies (color), and the effect size strength of the informative studies (x-axis). There were a total of 500 simulations at each setting.

Figure S11: Assessing the target FDR at level 0.01 using the oracle functional q -value, standard q -values, and functional q -values from sfFDR in the dependent SNP simulation study. The boxplot combines the “None,” “Moderate,” and “Large” effect size strength settings.

Figure S12: The empirical true discovery rate as a function of the empirical false discovery rate at target FDR level of 0.001, 0.002, . . . , 0.01 in the dependent SNP simulation study using the standard q -value (dark orange), functional q -value from sfFDR (green), and oracle functional q -value (black). We varied the power of the primary study (rows), the power of the informative studies (columns), and the effect size strength of the informative study (shape). Each point is the average from 500 replicates.

Figure S13: Comparing the functional p -value from sfFDR to various SNP selection procedures (color) that were used to train the model in the UK Biobank study. The informative traits were body fat percentage (BFP), cholesterol, and triglycerides. (a) The number of discoveries as a function of the proportion of the study sample size (i.e., downsampling proportion) at a significance threshold of 5×10^{-8} ; (b) the overlap in discoveries (or replication rate) with a meta-analysis approach.

Figure S14: The computational time to train sfFDR on a set of LD-independent SNPs as a function of the number of informative traits (color). The time to train sfFDR on the UKBB study and EGPA study are denoted by 'x,' which also includes the time to predict the left out LD-dependent SNPs. A single core of an Apple M3 processor was used in this analysis.

Figure S15: sfFDR applied to the BMI summary statistics in the UK Biobank leveraging BMI summary statistics from either FinnGen (left), GIANT (middle) or Million Veterans Program (MVP, European population; right) biobanks as an informative trait. (a) The number of discoveries as a function of the downsampling proportion. (b) The proportion of discoveries ($p < 5 \times 10^{-8}$) that overlapped with a meta-analysis of the UK Biobank data and either MVP (FinnGen and GIANT) or FinnGen (MVP). sfFDR was trained on 491,999, 461,811, and 166,416 overlapping SNPs with the FinnGen (sample size: 500,348), MVP (sample size: 424,231), and GIANT (sample size: 339,224) biobanks, respectively.

Figure S16: The functional proportion of null tests (top) and conditional density (bottom) from the UK Biobank analysis (downsampling proportion of 1) when (a) the primary trait is body mass index (BMI; denoted by p) and the informative trait is body fat percentage (BFP; denoted by z), (b) the primary trait is non-null BMI summary statistics and the informative trait is null BFP summary statistics, (c) the primary trait is null BMI summary statistics and the informative trait is non-null BFP summary statistics, and (d) the primary trait is simulated from a mixture of null and non-null BMI summary statistics and the informative trait is non-null BFP summary statistics. Note that only the null SNPs are shown in (d) and that each point is the average of 10 replicates (except at (a) and the top plot in (b)). The red dashed line denotes either the proportion of truly null tests (top plots) or the expected value (bottom plots).

Figure S17: Comparison of the estimated functional p -values from sfFDR and the estimated p -values from MTAG (color) compared to the UK Biobank standard p -values (x-axis) for BMI using a set of null correlated traits as informative studies (shape). There were 10 permutations of the null traits at each downsampling proportion and each point represents the average functional p -value across permutations. A log10 transformation was applied to both axes.

Figure S18: Quantile-Quantile plot of the p -values after applying the standard analysis, sfFDR, and MTAG to the UK Biobank study under a null setting where BMI is permuted and uncorrelated with the informative traits (shape). There were a total of 10 replicates at each setting.

Figure S19: The type I error rate at a significance threshold of 1×10^{-4} in the global null setting ($\pi_0 = 1$) using the standard p -values, functional p -values, and functional p -values with $\pi_0(z)$ set to be constant and the nearest neighbor parameter in the density estimator set to be 0.7. We varied the power of the informative studies (color). There were a total of 500 simulations at each setting.

Figure S20: Quantile-Quantile plot of the functional p -values from various SNP selection procedures to train sfFDR. (a) A set of known simulated null SNPs added to 19 random chromosomes in the primary trait (BMI) and not the informative traits (BFP, cholesterol, and triglycerides). (b) The primary trait (BMI) was permuted to be null and uncorrelated with the informative traits (BFP, cholesterol, and triglycerides), i.e., $\pi_0 = 1$. There were 10 replicates in each setting.

Figure S21: sfFDR applied to BMI using the informative traits body fat percentage (BFP), cholesterol (CHO), triglycerides (TRI), and a number of null (or uninformative) traits (columns). The traits were fitted in sequence (linetype) where a set of uninformative traits (columns) were added to assess the performance under a mixture of informative and uninformative traits. (a) The number of discoveries as a function of the downsampling proportion. (b) The overlap in discoveries (or replication rate) with a meta-analysis approach.

Figure S22: Comparison of functional p -values from sfFDR to standard p -values for the EGPA study when the informative traits are the UK Biobank null traits. Each point is the average of apply sfFDR to 10 replicates. A log10 transformation was applied to both axes.

Figure S23: Comparison of functional p -values from sfFDR to standard p -values for the EGPA study when the informative traits are the UK Biobank obesity-related traits. A log10 transformation was applied to both axes.

Figure S24: Fine mapping results using the functional local FDR estimates from sfFDR. For each lead SNP, the 95% credible set (CS) is shown in red for EGPA including SNPs 500kb upstream and downstream of the lead SNPs. The top plot in each set shows the local Manhattan plot while the bottom plot shows the fine mapping posterior probabilities calculated under the assumption of a single causal variant. We distinguish the SNPs in the CS that also overlap with the CS from ASTAO (square), ASTCO (triangle), and EOSC (diamond).

Figure S25: Comparison of the published p -values from the EGPA study and the p -values used in sfFDR.

Figure S26: Comparing the RMSE of the estimated proportion of truly null tests from the standard q -value, sfFDR, Adapt, CAMT, and Boca-Leek in the global null independent SNP setting. There were a total of 500 replicates at each combination of primary study power (rows), informative study power (columns), and the effect size strength of the informative studies (color).

Figure S27: Comparing the estimated proportion of truly null tests from the standard q -value, sfFDR, Adapt, CAMT, and Boca-Leek in the global null independent SNP setting. There were a total of 500 replicates at each combination of primary study power (rows), informative study power (columns), and the effect size strength of the informative studies (color).